# A general fruit acid chelation route for eco-friendly and ambient 3D printing of metals

Soo Young Cho [1,7], Dong Hae Ho[1,7], Yoon Young Choi[1], Soomook Lim[2], Sungjoo Lee [3,4], Ji Won Suk[2,4,5], Sae Byeok Jo [6✉] & Jeong Ho Cho [1✉]

Recent advances in metal additive manufacturing (AM) have provided new opportunities for prompt designs of prototypes and facile personalization of products befitting the fourth industrial revolution. In this regard, its feasibility of becoming a green technology, which is not an inherent aspect of AM, is gaining more interests. A particular interest in adapting and understanding of eco-friendly ingredients can set its important groundworks. Here, we demonstrate a water-based solid-phase binding agent suitable for binder jetting 3D printing of metals. Sodium salts of common fruit acid chelators form stable metal-chelate bridges between metal particles, enabling elaborate 3D printing of metals with improved strengths. Even further reductions in the porosity between the metal particles are possible through post-treatments. A compatibility of this chelation chemistry with variety of metals is also demonstrated. The proposed mechanism for metal 3D printing can open up new avenues for consumer-level personalized 3D printing of metals.

[1] Department of Chemical and Biomolecular Engineering, Yonsei University, Seoul 03722, Republic of Korea. [2] School of Mechanical Engineering, Sungkyunkwan University, Suwon, Gyeonggi-do 16419, Republic of Korea. [3] Department of Nano Engineering, Sungkyunkwan University, Suwon, Gyeonggi-do 16419, Republic of Korea. [4] SKKU Advanced Institute of Nanotechnology (SAINT), Sungkyunkwan University (SKKU), Suwon, Gyeonggi-do 16419, Republic of Korea. [5] Department of Smart Fab. Technology, Sungkyunkwan University, Suwon, Gyeonggi-do 16419, Republic of Korea. [6] School of Chemical Engineering, Sungkyunkwan University, Suwon, Gyeonggi-do 16419, Republic of Korea. [7] These authors contributed equally: Soo Young Cho, Dong Hae Ho. ✉email: eos0523@gmail.com; jhcho94@yonsei.ac.kr

Additive manufacturing (AM), also known as three-dimensional (3D) printing, is one of the most important technologies in the fourth industrial revolution because it can enable the personalization of products and rapid prototyping. In an attempt to expand the boundaries of AM, numerous researchers have focused on developing printable materials[1–3] and corresponding techniques for 3D printing[4–7]. Consequently developed advanced and sophisticated printable materials and 3D printing techniques have accelerated the utilization of AM in various industries, such as aerospace[8,9], biomedicine[10–12], and the food industry[13,14]. Research on metal AM that can facilitate its application to various industrial fields has also been actively conducted[15–18]. However, unlike polymer AM, metal AM still has applicability only at the industrial and academic scales because of the demanding conditions of the printing environment, which hinder realization of consumer-level desktop applications. Selective laser melting and electron beam melting have been proposed as breakthrough technologies[19–21], but printing processes requiring high-power energy sources, inert gas atmospheres, and high-temperature preheating have limited their application range[22].

Binder jetting metal 3D printing (BJM3DP) is a promising AM technique that selectively jets a liquid binding agent onto metal powder, which results in bond formation between particles[23–25]. The technological challenges for the commercial adaptation of BJM3DP still involve overcoming demanding conditions of metal AM processes including materials handling, post-treatments, and quality control. However, BJM3DP has particular advantages over other metal 3DPs, stemming from its high feasibility toward operationally low-cost, simple, and safe 3DP processes[26]. The ambient conditions of initial printing process[27–30], as well as the possible use of commercially available ink cartridges constitute high accessibility to this technology than others, which can potentially facilitate consumer-level desktop applications. Moreover, as a groundwork for BJM3DP to become a more accessible green technology adequate for both industrial and personalized uses such as rapid prototyping, one of the important aspects to be explored is adapting and understanding of environmentally friendly ingredients including binder materials. The two most commonly used hazardous binding agents, 2-butoxyethanol-based solution and 2-pyrrolidone-based solution, have been specifically considered to be responsible for such issues[31,32]. Additionally, the recently developed metal-organic dispersion ink composed of cupric formate and octylamine has also been found to have an adverse environmental impact[33,34]. In the field of ceramic BJ3DPs, especially for biomedical applications, there already are various investigations regarding the use of non-hazardous binder materials such as green-solvent-soluble polymers, maltodextrin, sugar, and corn starch[35–37]. However, only a few candidates for metal BJ3DP have been explored so far, and the reported characteristics of printed objects such as the porosity and mechanical strengths are far below the ones based on the aforementioned common binder materials[38]. Therefore, it is now imperative to broaden the technological horizon thorough developing new green binding agents for metals that can be eco-friendly as well as non-hazardous[24,39–41], with prospects of simultaneously achieving the desired properties of printed objects.

Here, we introduce a binding mechanism for BJ3MDP that is based on the use of a chelator composed of salts of naturally available fruit acids as an eco-friendly binding agent. Metal-chelate bridges between metal particles are successfully formed *via* ink-jetting of water onto a powder composed of a uniform mixture of the metal particles and chelator. This metal-organic complexing mechanism is thoroughly analyzed by Fourier transform infrared (FT-IR) spectroscopy, X-ray photoelectron spectroscopy (XPS), and scanning electron microscopy (SEM). Then, compression tests are performed on the metal 3D-printed object to confirm the dependence of its mechanical strength on the type of chelator. Subsequently, the mechanical strength is further improved by post-treatments as well as optimizing the distribution of particle sizes and compositions. Finally, objects of various shapes are printed using various metals, which demonstrates that the proposed chelator-assisted BJM3DP technique is not only useful for the realization of complex and sophisticated architectures but also applicable to a wide range of metal powders. The environmentally friendly chelator presented herein is expected to promote greener metal 3D printers adequate for both industrial and consumer-level scale applications.

## Results

**Eco-friendly BJM3DP**. Figure 1a shows the 3D scheme of the BJM3DP system, which has four main components: an inkjet cartridge, powder gantry boxes, a roller, and x- and y-axes stages. Prior to the 3D printing process, the powder and inkjet cartridge were prepared as follows (see Supplementary Fig. 1 for photographic images of each component). First, the metal powder was mixed with the chelator. Here, the nozzle clogging issue from the precipitation of binding agents inside inkjet cartridge was avoided since the chelator functions as a solid-phase binding agent pre-mixed with metal particles, which also is one of the commonly adopted approaches in biomedical fields[35–37,42]. Next, two gantry boxes were filled with the mixture of the metal powder and chelator; for the purpose of this work, Al powder was utilized. Each of these gantry boxes has a different purpose. One is the builder gantry box in which objects are 3D-printed, and the other is the feeder gantry box that stores and supplies the powder to the builder gantry box. Then, the inkjet cartridge was filled with deionized (DI) water, which activated the chelation reaction. After the preparation process, the printing cycle was initiated through the deposition of a powder layer on top of the builder gantry box. During the printing process, the builder platform moved one step downward to provide space for one powder layer and the feeder platform simultaneously moved one step upward to push up the powder. Then, the roller module positioned at the feeder gantry box moved toward the builder gantry box to supply powder and flatten powder protruding on top of the latter box, as shown in Fig. 1b. Once a powder layer was deposited, the module returned to its original position. Subsequently, DI water was jetted from the inkjet cartridge onto the powder layer deposited on top of the builder gantry box at programmed positions (Fig. 1c). This cyclic process was repeated until the uppermost layer of the designed 3D object was deposited. Upon completion of the printing cycle, the 3D-printed object was removed from the pile of metal powder and the unchelated powder was subsequently removed through air blowing. Photographic images of the overall metal 3D printing process are shown in Supplementary Fig. 2. As a safety disclaimer, the Al powder is classified with ST3 explosive ratings, which can pose a potential danger for Al powder-based systems. Furthermore, a plausible reaction between water and Al could also generate combustible hydrogen during the process, thus a proper ventilation and an equipment of hydrogen detectors are required for the safe operation. In our experimental system (capable of jetting 250 ml of water over 6 h), 193 ppm/hour of hydrogen generation was observed, which would indeed pose danger over 200 h of continuous printing in a completely closed system by reaching the lower explosive limit of the hydrogen (41,000 ppm)[43].

As binding agents, nature-based chelators, which are crucial to ensure eco-friendliness of metal 3D printing, were used. Unlike polymer binding agents, which are hazardous, the chelators

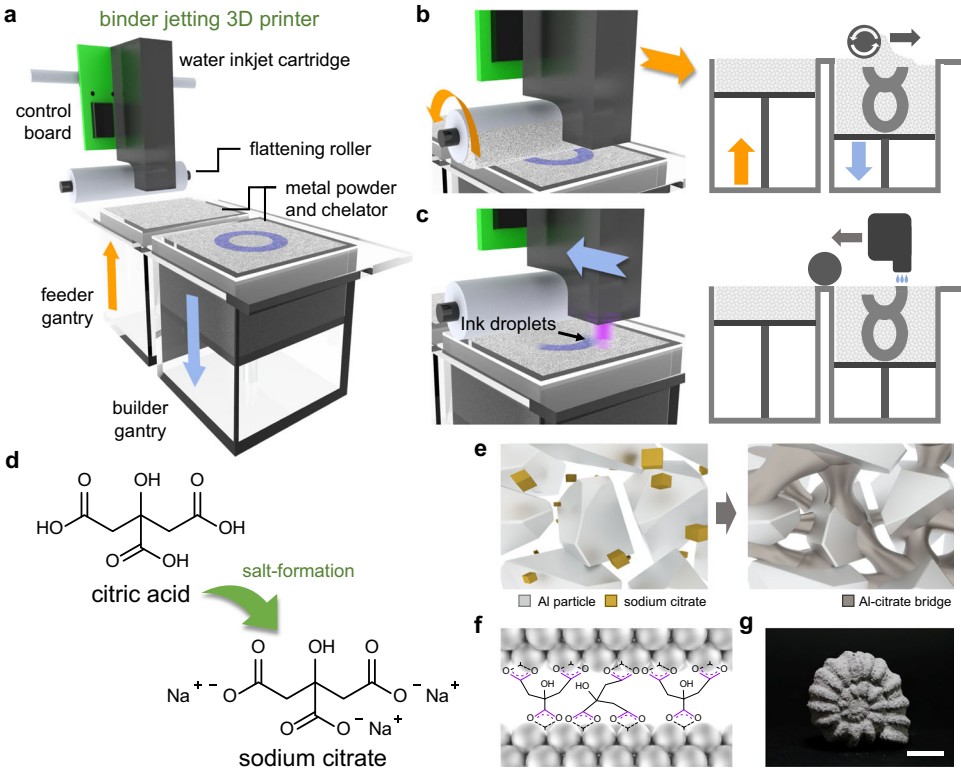

**Fig. 1 Eco-friendly BJM3DP. a** Schematic illustration of BJM3DP system. **b** Powder layer deposition on builder gantry box. **c** Programmed water ink-jetting onto powder layer. **d** Chemical structures of naturally available eco-friendly fruit acid (top) and its sodium salt (bottom). **e** Schematic description of as-prepared mixture of metal powder and chelator (left) and metal-chelate bridge formation after metal chelation (right). **f** Chemical structure of metal-chelate bridges on metal particle surface. **g** Photographic image of 3D-printed ammonite-shell-shaped object (scale bar: 10 mm).

utilized here are food-grade organic materials because they are salts of nature-based fruit acids (e.g., of fruits such as lemons, cherries, and grapes)[44]. For example, citric acid derived from citrus fruits has three carboxyl groups, and it transforms into sodium citrate upon replacement of the hydrogens in these groups with sodium ions through a salt-formation reaction (Fig. 1d)[45]. Here, the carboxyl group of sodium citrate plays an important role in the metal chelation reaction. Upon wetting of the uniform mixture of the metal powder and chelator (Fig. 1e), metal chelation occurs on the surface of the metal particles, which induces the formation of metal-chelate bridges between the particles. Figure 1f and Supplementary Fig. 3 show the chemical structure of metal-chelate bridges formed on metal particle surface and the change in the microstructure of Al powder. Successful chelation imparts structural integrity to the 3D-printed object and its architecture is consequently maintained, as a result of which the object has a precise and sophisticated shape, as shown in Fig. 1g.

**Formation of metal-chelate bridges between metal particles.** Figure 2a shows the mechanism of chelate complex formation between Al particles and the underlying chemical reaction. When water droplets are jetted onto Al powder, water permeates between the particles and gradually dissolves the chelator. Then, the ionized chelator solution preferentially attacks intrinsic defects in the Al particles, consequently producing Al-chelate compounds, which results in the bridging of the Al particles. Figure 2b shows FT-IR spectra of Al objects printed using the following proposed chelators: sodium salts of citrate (NaCit), tartrate (NaTar), succinate (NaSuc), and ascorbate (NaAsc). These four chelators have three, two, two, and one coordinate donor sites, respectively, and the

difference in the number of coordination sites affects the formation of metal-chelate bridges. In the spectra of the carboxyl-based chelators (NaCit, NaTar, and NaSuc), two major bands, which originate from the asymmetric stretching vibration ($\nu_{as}(COO^-)$) and symmetric stretching vibration ($\nu_{s}(COO^-)$) of the carboxylate group ($COO^-$), are present in the frequency ranges of 1540–1720 $cm^{-1}$ and 1320–1470 $cm^{-1}$, respectively[46]. In the spectra of NaCit, $\nu_{as}(COO^-)$ at 1592 $cm^{-1}$ and $\nu_{s}(COO^-)$ at 1393 $cm^{-1}$ are blue-shifted by 4 $cm^{-1}$ and red-shifted by 11 $cm^{-1}$, respectively, after chelation. The increase in the molecular mass due to the chelation of the carboxylate group and Al causes a change in the vibration frequency, which results in a band shift[47]. The peak shifts of $\nu_{as}(COO^-)$ and $\nu_{s}(COO^-)$ thereby indicate the formation of the chelate complex on the Al particle surface (the FT-IR spectrum of pristine Al powder in the same frequency range is shown in Supplementary Fig. 4). The separation of peaks ($\Delta\nu = \nu_{as} - \nu_{s}$) was further analyzed to identify the difference in the types of coordination between Al and the carboxyl-based chelators[48]. Because NaCit, NaTar, and NaSuc chelated with Al have a lower value of $\Delta\nu(COO^-)$ than do the chelators themselves (Supplementary Table 1), these chelators coordinate with Al in the bidentate chelating form, which enables a single metal atom to have two bonds with a carboxylate group, as illustrated in the left panel of Fig. 2b. Unlike with these carboxylate-based chelators, NaAsc forms only a single bond with the Al atom, as indicated by the non-split $\nu_{C=O}$, the band broadening of $\nu_{C-O}$, and the decreased intensity of the hydroxyl group after chelation[49].

The XPS spectra of chelated Al provide information about the extents of chelate complex formation when the different chelators are used. The left panel of Fig. 2c and Supplementary Fig. 5 show the Al 2$p$ and O 1$s$ spectra of the chelated and pristine Al powders, respectively. The deconvoluted Al 2$p$

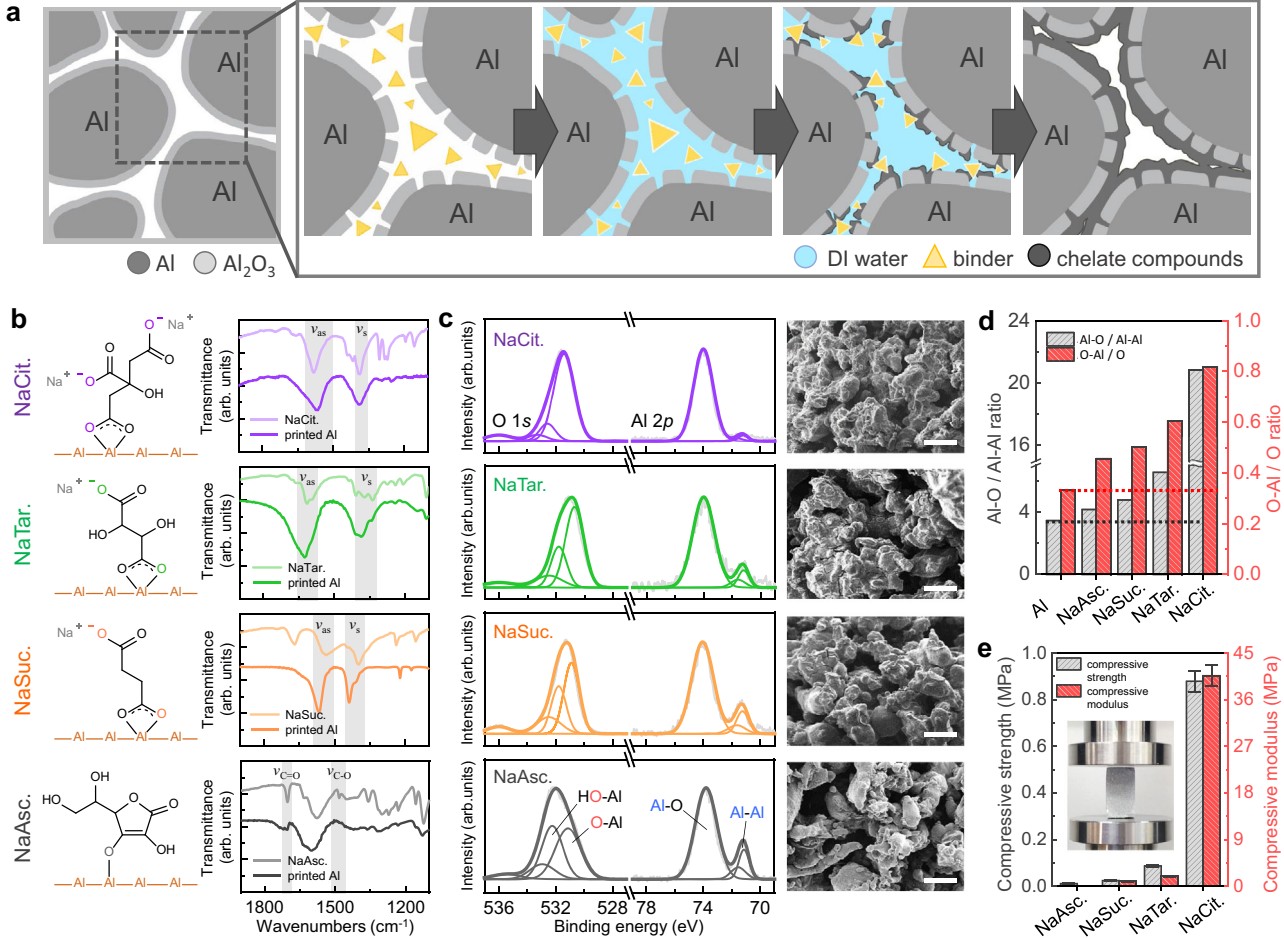

**Fig. 2 Formation of metal-chelate bridges between metal particles. a** Schematic illustration of mechanism of metal chelation on metal particle surface.
**b** Chemical structures of chelators and their coordination with metal particle surface (left) and FT-IR analysis with identified vibration peaks (right).
**c** Deconvoluted XPS peaks (left) and cross-sectional SEM images of chelated metal particle surface (right) (scale bar: 100 μm). **d** Al–O/Al–Al atomic ratio
(integrated areas of peaks in Al 2p spectra) and O–Al/O atomic ratio (integrated areas of peaks in O 1s spectra) for various chelators. **e** Compressive
strength and modulus of objects 3D-printed by BJM3DP using various chelators. Data are presented as mean values ± standard deviations. The inset shows
a photographic image of compression testing of the 3D-printed object.

spectra have three distinct components: Al $2p_{3/2}$ (71.6 eV), Al
$2p_{1/2}$ (72.0 eV), and Al–O (74.2 eV). The Al–O peak originates
from the oxides and hydroxides formed on the Al surface[50]. As
depicted in Fig. 2d, the four types of chelated Al show a larger
Al–O/Al–Al atomic ratio than the pristine Al particles.
Furthermore, the atomic ratio increases as the number of
coordination bonds increases. These results indicate that the
chelators play a pivotal role in Al–O formation, and greater the
number of coordination bonds, higher is the thickness of the Al–
O layers formed on the Al particle surface, as observed in the
SEM images (right panel of Fig. 2c). This tendency of formation
of a thicker Al–O layer is also confirmed from the O 1s spectra.
The O 1s peak has three components, which correspond to O–
Al, HO–Al, and the chemisorbed water and are positioned at
530.7, 531.8, and 532.4 eV, respectively[51]. As can be seen from
the plot in Fig. 2d, the relative amount of O–Al increases as the
number of coordination bonds increases; this trend reveals that
a chelator with more coordination sites is favorable for the
formation of the Al-chelate complex. The promoted chelation
results in stronger bonding between the metal particles, and
therefore, increases the mechanical strength of the 3D-printed
objects (Fig. 2e). The object printed using NaCit has the highest
compressive strength (0.88 MPa) and compressive modulus
(40.66 MPa). The improved mechanical properties of 3D-

printed object based on NaCit than other chelators in our study
can be explained by the fact that coordinate donor sites
contribute to the strength of the 3D-printed object. Changes in
the atomic ratio and mechanical properties with the type of
chelator because of the difference in the number of coordination
groups are depicted in Supplementary Fig. 6. NaCl was
subsequently added to release more metal ions from the
intrinsic defects in the Al particles and increase the binding
strength of the printed object. Upon the addition of NaCl, the
chloride ions attacked the Al defects to form Al ions, which, in
turn, promoted chelation on the surface of the metal particles in
an aqueous environment (Supplementary Fig. 7)[52]. The
enhancement of chelation through NaCl addition was also
confirmed by SEM, FT-IR spectroscopy, and XPS (Supplemen-
tary Figs. 8 and 9), all of which revealed numerous metal-chelate
bridges. The increase in the density of metal-chelate bridges, in
turn, led to an increase in the compressive modulus to
68.36 MPa; this is a 1.7-fold increase compared to that of the
3D-printed object not treated with the NaCl additive, as shown
in Supplementary Fig. 10. Further enhancements in the
mechanical strengths can very well be acquired through post-
treatments as well as optimizing the distribution of particle sizes
and compositions (Supplementary Fig. 11), which are described
in the following section.

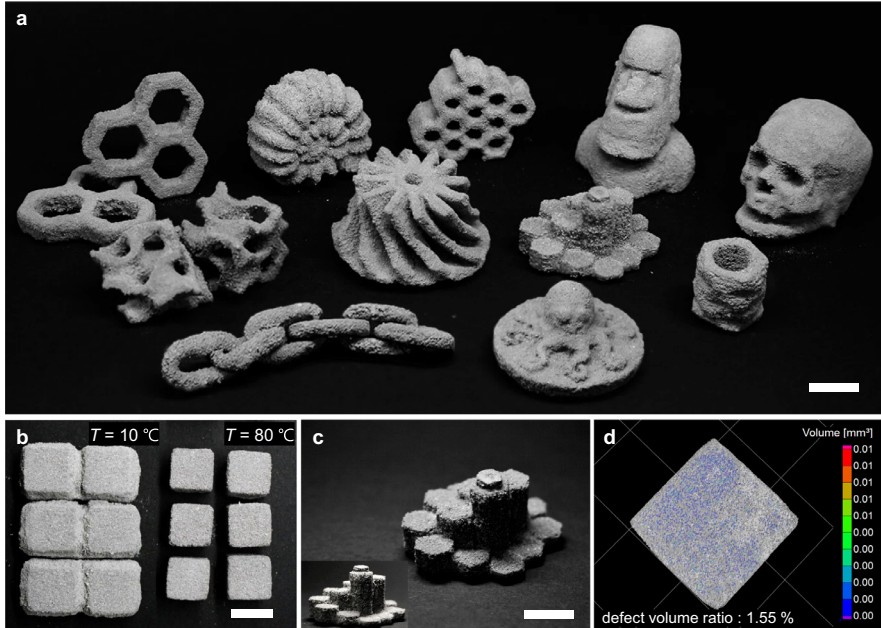

**Fig. 3 Various objects 3D-printed using NaCit chelator.** Photographic images of **a** 3D-printed Al objects having various shapes. **b** Demonstration of the minimization process for bleeding effects by using different initial printing conditions. The temperature of builder was gradually increased from 10 to 80 °C. **c** Photographic images of representative 3D-printed objects, which depict their high dimensional accuracy. **d** Rendered image and defect volume ratio from μCT analyses for green body based on metal particles with a bimodal powder distribution based on 10 and 75 μm metal particles. The scale bar on the images is 10 mm. Three-dimensional model data for the ammonite shell (thing: 1611970), moai statue (thing: 3905999), octopus (thing: 159217), gyroid cube (thing: 757884), skull (thing: 11953), hexagon stair sculpture (thing: 46966), and faceted cup (thing: 414252) were acquired from the open-source website www.thingverse.com. Other 3D models were designed by the authors using Fusion 360 software.

**Applicability of NaCit chelator for 3DP of metal objects**. To demonstrate the capability of our developed BJM3DP system in producing elaborate objects, we designed and 3D-printed various 3D structures, as shown in Fig. 3a and Supplementary Fig. 12. For the industrial applications of BJM3DPs, various aspects should be taken into accounts, including the bleeding effect, stair-stepping effect, liquid-powder imbibition effect, as well as sensitive processing parameters such as powder size distribution, powder layer consistency, and layer shifting during drying[53,54]. Those aspects critically determine the resolution and mechanical strengths of 3DP objects. As a first step, testing and securing the microstructural integrity of 3D printed objects were considered in the fabrication of objects with sophisticated geometries, such as a gyroid cell, an impeller, an ammonite shell, and a skull. In addition, the bleeding issue—a phenomenon in which the binding agent solution adversely flows out of the printed object due to the capillary forces—was addressed to ensure high printing quality[24]. The use of water as a jetting fluid for the initial printing process as well as a for the post-treatment enabled a partial mitigation of this issue: it has higher vapor pressure than conventional binding agent solutions, because of which it evaporates faster and therefore reduces the solution from flowing out of the prescribed structure, especially at elevated printing temperatures used in our study (Fig. 3b and Supplementary Fig. 14). A post-treatment step of the humidification then ensures the completion of the remnant chelation reaction between metal particles without causing adversary bleeding (discussed in the following section). Consequently, the dimensional accuracy of the 3D-printed objects as shown in Fig. 3c can be acquired. Figure 3d and Supplementary Fig. 15 show the rendered images and extracted void ratios of the printed object from computed 3D X-ray microtomography (3D μCT) analyses. A good packing density (defect volume ratio of 4.92%) of the green body based on coarse Al powder (<355 μm) by itself not only enables the 3D object to retain its shape but also promotes the solidification of the structure during the

subsequent post-treatment process, thereby providing structural integrity. On the other hand, to illustrate prospects of further enhancing the mechanical strengths of BJM3DP objects, we optimized the distribution of the powder sizes and compositions[55,56]. In regards to overcoming the low intrinsic porosity of BJM3DP objects (30–60% in general) and reducing microscopic void ratios, we used a bimodal powder distribution based on 10 and 75 μm metal particles (Supplementary Fig. 16). The improved packing of the metal powders from the filling of interstitial voids with fine particles (Supplementary Figs. 15 and 17) thus led defect volume ratios to be critically reduced (1.55 and 0.30% for green and sintered body, respectively). As a result, greatly improved mechanical strengths and compressive modulus of green body up to 6.06 MPa and 218.43 MPa, respectively, were obtained. After completing thermal debinding and sintering processes for printed green bodies (Fig. 4a), the values could further be improved to 29.59 MPa and 1.49 GPa, respectively (Fig. 4b). These results are indeed much superior than the values acquired in previous reports on green-solvent-soluble binders, and are even comparable to the Al (or Al alloy)-based metal objects printed by other much more sophisticated 3DP methods based on high powered lasers (Fig. 4c and Supplementary Fig. 18).

**Post-treatment process and several 3D-printable metals**. The post-treatment process performed for strength improvement in the standard metal BJM3DP technique can also be applied to the chelation-based metal 3D printing technique proposed herein. The post-treatment consists of a two-step process: humidification followed by thermal sintering. First, the unreacted chelator between particles was fully reacted through humidification (see Supplementary Fig. 19 for the SEM image of the humidified 3D-printed object). Since we chose to use a marginal water droplet size (33 pL) at elevated evaporation conditions (builder/nozzle temperature of 70 °C) to minimize the bleeding effect (see

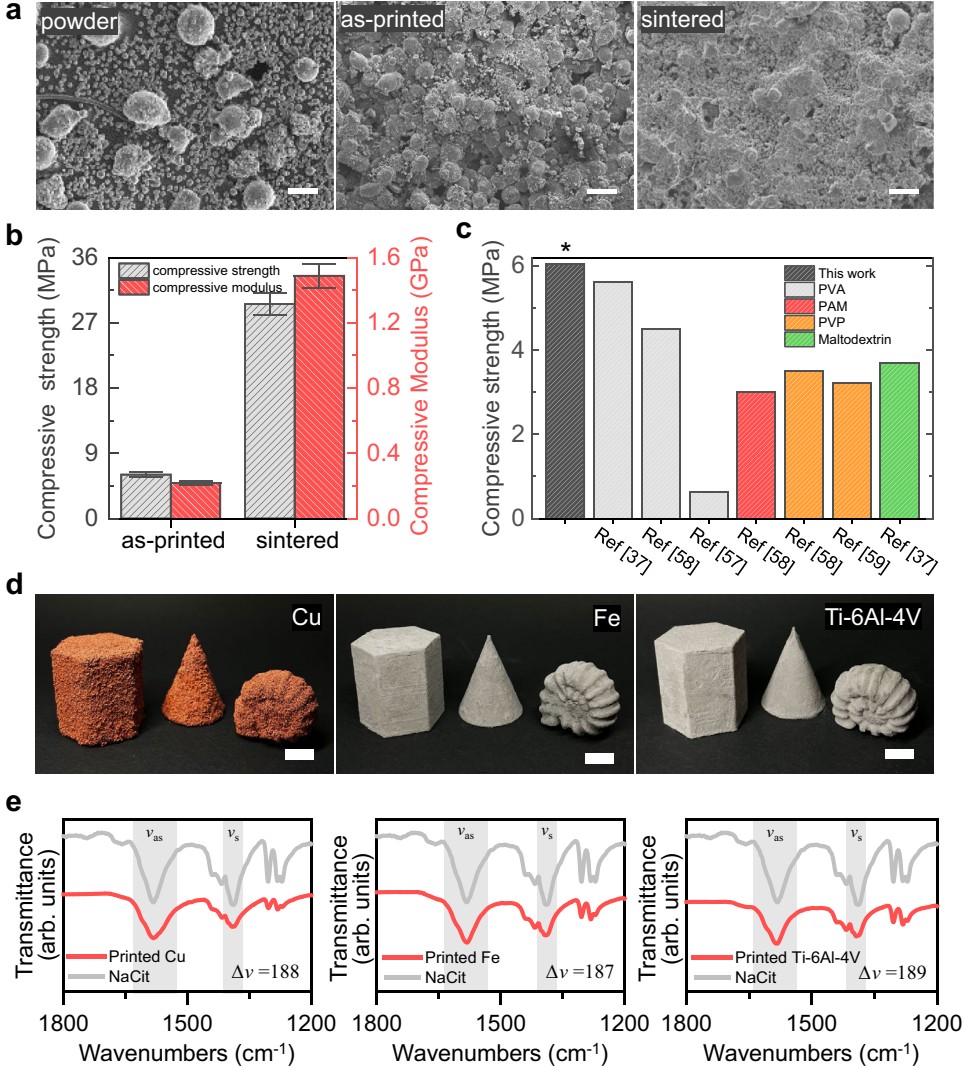

**Fig. 4 Post-treatment process and several 3D-printable metals. a** Cross-sectional SEM images of powder (left), as-printed 3D object (middle) and sintered 3D object (right) (scale bar: 100 μm). **b** Comparison of compressive strength and modulus of as-printed 3D object and sintered 3D object. Data are presented as mean values ± standard deviations. **c** The recent technological progress in BJM3DP techniques, with respect to the mechanical strengths of printed green bodies based on various binder materials. **d** Photographic images of objects 3D-printed using Cu (left), Fe (middle), and Ti–6Al–4 V alloy (right) (scale bar: 10 mm). **e** The FTIR spectra and corresponding peak separation ($\Delta\nu$(COO$^-$)) of 3D printed Cu, Fe, and Ti−6Al−4V objects compared to the one for pure NaCit.

Supplementary Fig. 14 for details), an additional step of the humidification can complete the reaction between surplus chelator and particles, and indeed boosts the strengths of the printed objects without sacrificing the printing resolution. Subsequent thermal debinding and sintering (see Supplementary Fig. 20 for detailed procedure) caused solid-state diffusion of metal particles, which led to a decrease in the interparticle porosity, as shown in Supplementary Fig. 21. In other words, the metal particles came into closer contact with their neighboring particles, as shown in Fig. 4a. Consequently, the compressive strength and compressive modulus of the thermally sintered 3D-printed objects are greatly improved, compared to those in the as-printed state (Fig. 4b).

These strategies are expected to contribute to improvement in the mechanical properties of printed objects through minimization of the void ratios and porosity in final products. Figure 4d shows objects 3D-printed using copper (Cu), iron (Fe), and a titanium–aluminum–vanadium alloy (Ti–6Al–4V). The printed objects showed structural integrity that cannot be acquired through jetting of water alone without our solid-phase binder

mixed with metal particles (Supplementary Fig. 22). Furthermore, FT-IR spectra of all printed metal objects exhibit noticeable shifts in $\Delta\nu$(COO$^-$) (188 cm$^{-1}$ for Cu, 187 cm$^{-1}$ for Fe and 189 cm$^{-1}$ for Ti−6Al−4V) compared to pure NaCit (198 cm$^{-1}$), which agree with the trend we observed in Al-based printed objects (Fig. 4e). Changes in ionic compositions between pure and BJM3DP-printed metals were also observed, wherein Cu$^{2+}$, Fe$^{3+}$, and Al$^{3+}$ can preferentially chelate with COO$^-$ of citrate ions (Supplementary Fig. 23). These results demonstrate that our chelator is feasible for forming chelation bridges between metal particles of other metals, and thus can be applied as a binder for BJM3DPs of various metals and alloys than just Al alone.

## Discussion

An eco-friendly chelator-based binding mechanism for BJM3DP was proposed and the feasibility of achieving an effective 3D printing system was demonstrated in this study. In the field of BJM3DPs, a few candidates for green-solvent-soluble binders have been reported so far, such as polyvinyl alcohol (PVA)[37,42,57], polyacryl amide

(PAM)[58], polyvinyl pyrrolidone (PVP)[59], and maltodextrin[37], some of which were adopted from successful demonstrations in the field of ceramic BJ3DPs for biomedical applications[58]. However, the mechanical properties of the printed metal objects based on them, even with highly tough Ti−6Al−4V alloy, were mostly rather less competitive for broad applications compared to the ones based on common binder materials developed without considerations for eco-friendliness. Among them, dextrin, a liquid-based binder, has shown promises that can provide meaningful mechanical properties to the printed green bodies of Al/alumina (Fig. 4c). As a solid-phase binder, PVA has been adopted but the mechanical properties achieved was abysmal[57] contrary to previous demonstrations of reasonable binding properties in ceramic (hydroxyapatite)[37] BJ3DPs. In comparison, results in this work demonstrated that the citric acid, as an another solid-phase green binder that only requires the jetting of pure water for the structural formation, can endorse prospects of simultaneously achieving desired mechanical properties of printed objects as well as the eco-friendly and facile printing processing. By using citric acids as binders, the structural integrity of the 3D-printed objects can be achieved through the efficient formation of metal-chelate bridges between metal particles. A combination of optimized powder size distribution and the efficient binding properties of citric acid, therefore, enabled high packing densities for both green and sintered body of printed Al (1.55 and 0.30% defect volume ratios, respectively). The optimized packing then led to the achieved good mechanical strengths (~30 MPa in compressive strength) for Al objects, which were comparable to or even higher than the ones printed by other sophisticated metal 3DPs based on high powered lasers (Supplementary Fig. 18). The porosity of the green body, on the other hand, still has a room for enhancements, even though the value (41%) itself is well within the previous reports (30–60%)[60–62]. Furthermore, the resolution of the 3D printed objects, which is another important aspect along with mechanical strengths, can be enhanced by controlling the evaporation of jetted water. The inevitable liquid-powder imbibition could be reduced by elevating the builder temperature near evaporation point (70 °C) of water, of which the plausible insufficiency in the metal bridging from the fast evaporation was compensated by using an additional step of the humification that completes the remnant chelation process between metal particles and citrates (Supplementary Fig. 19). The whole printing process based on citric-acid-based BJM3DP can be completed through the thermal treatments of green bodies, where the debinding and sintering result in the merging of metal particles and thus provides an abrupt enhancement in the mechanical strengths. In order to minimize the rapid shrinkage and maintain the structural integrity during the initial debinding process, the optimization of debinding temperature and duration is required to remove an appropriate amount of chelation bridges at mild rate. Since the citric acid thermally decomposes above 175 °C, which is far lower than that of other water-soluble polymeric binders such as PVP, it is necessary to choose lower debinding temperature. A fine adjustments and optimizations are out-of-scope for this work and would require further study, but our choice of conditions (350 °C for 3 h) did not sacrifice the structural integrity. This work also demonstrated that, the chelation chemistry of citric acid observed in Al could be expanded to various metals such as Cu, Fe, and Ti−Al6−4V alloy. Based on these aspects, the proposed facile approach of using environmentally friendly chelators is expected be a cornerstone for promoting the development of the highly approachable, low-cost, and safe consumer-level desktop metal 3D printing systems.

## Methods

**Materials**. The four chelator powders were prepared by grinding sodium ascorbate (≥99%, Sigma-Aldrich), sodium succinate (≥99%, Sigma-Aldrich), sodium tartrate (≥99%, Sigma-Aldrich), and sodium citrate (≥99%, Sigma-Aldrich). Al powder was purchased from Henan Yuanyang Powder Technology Co., Ltd (FLPN10 for 10 μm),

from Korea Powder Co.,(CAS#: 7429-90-5 for 75 μm powders) and from Changsung (≤40 mesh, 355 μm). The size distribution was confirmed by using ISO 13320 laser diffraction method (Beckman Coulter LS-13-320), as presented in Supplementary Fig. 16. Then, each chelator powder was uniformly mixed at a concentration of 20% (w/w) with Al powder (≤40 mesh, Changsung) and then sieved through a mesh. Sodium chloride (≥99%, Sigma-Aldrich) with a concentration of 5% (w/w) was used as an additive. The preparation procedure of the other three metal powders (Cu (≤40 mesh, Sigma-Aldrich), Fe (≤325 mesh, Duksan), and Ti–6Al–4 V (≤325 mesh, Grade 5, Korea Powder)) was the same as that of the metal chelator mixture.

**BJM3DP process**. The desktop printer for BJM3DP was assembled using a fusion deposition modeling 3D printer kit (Geeetech I3 Pro, Shenzhen Getech Co., Ltd.). An acrylic gantry box add-on set (Colorpod, Spitstec, Netherlands) was mounted on the assembled metal 3D printer. BJM3DP was performed in an environment in which room temperature (~25 °C) and 20% relative humidity were maintained. An inkjet cartridge (HP45, Hewlett-Packard) was filled with 30 mL of DI water. The printing speeds was fixed at 2000 mm min$^{-1}$. The temperature of both the builder platform and the inkjet nozzle was maintained at 70 °C to control the evaporation rate of water during the 3D printing process. The process parameters are summarized in the Supplementary Table 2. After the 3D printing process, the printed object was humidified by exposing it to a humidifier spraying water at a rate of 35 mL h$^{-1}$ in a cylindrical chamber for 30 min. Subsequently, the printed object was thermally sintered in a tube furnace (LHA-12/300, Lenton) under vacuum (0.03 torr). The heat treatment profiles are shown in Supplementary Fig. 20. 3 cycles of the following heat treatment profile are conducted: (Debinding) Heating from RT to 350 °C by 10 °C/min followed by 350 °C treatment for 3 h. (Sintering) Heating from 350 to 620 °C by 10 °C/min followed by 620 °C treatment for 15 h. (Cooling) Cooling from 620 °C to RT by −10 °C/min.

**Characterization**. The formation of a metal-chelate complex between the Al particles in the 3D-printed objects was confirmed by FT-IR spectroscopy (VERTEX 70, Bruker Corporation, Germany) and XPS analysis (ESCALAB 250Xi, Thermo-Scientific, USA). The 3D-printed objects were visualized by SEM (JEOL-7800F, JEOL, Ltd., Japan). For compression tests, objects were 3D-printed according to the ASTM E9 standard. Compression tests were performed using a universal testing machine (QC-506M2F, Cometech) with a compression rate of 8 mm s$^{-1}$. Data are presented with mean values ± standard deviations (s.d.). The bulk density and porosity of printed objects were acquired by using 3D computed tomography (μCT, Nikon XTH 320) with the voxel size of 1 μm at X-ray beam energy of 210 kV. Voxel analysis and 3D visualization were performed by using VGSTUDIO MAX (Volume Graphics Pte. Ltd). The inter-particle porosity of the 3D-printed objects was measured by mercury intrusion porosimetry (PM33GT, Quantachrome).

## Data availability

All data generated or analyzed during this study are included in this published article (and its Supplementary Information files).

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

## Acknowledgements

J.H.C. was supported by the Creative Materials Discovery Program (2019M3D1A1078299) through the National Research Foundation (NRF) of Korea funded by the Ministry of Science and ICT, Korea, the Basic Science Program through the NRF of Korea funded by the Ministry of Science and ICT, Korea (2020R1A4A2002806), and the Korea Medical Device Development Fund grant funded by the Korea government (the Ministry of Science and ICT, the Ministry of Trade, Industry and Energy, the Ministry of Health & Welfare, the Ministry of Food and Drug Safety) (Project Number: KMDF_PR_20200901_0093, 9991006766). J.H.C. was partially supported by the Yonsei Signature Research Cluster Program of 2021 (2021-22-0004).

## Author contributions

J.H.C. conceived the concept, designed all the experiments, and supervised the work. S.Y.C. and D.H.H. carried out most of the experimental work and data analysis. Y.Y.C. assisted the data analysis with all other authors. S.Li, S.Le, and J.W.S. performed the mechanical property analysis of 3D printed object. S.B.J. led and supervised the whole revision process, including designing experiments, reorganizing the rationale, and editing the manuscript. All authors discussed the progress of research and contributed to editing the manuscript.

## Competing interests

The authors declare no competing interests.
