## [Peer Review File · Nature Communications]

Title: A General Fruit Acid Chelation Route for Eco-friendly and Ambient 3D Printing of MetalsREVIEWER COMMENTS

Reviewer #1 (Remarks to the Author):

The manuscript “A General Fruit Acid Chelation Route for Eco-Friendly and Ambient 3D printing of metals” presents a technique to 3D print metals such as Al, Cu, & Fe while avoiding hazardous binding agents and possibly providing a route to commercializing 3D metal printing. The manuscript will be of interest to a technical as well as academic audience. Taken as a whole, the manuscript is well written and easy to read.

A few questions that need to be considered include:

Lines 154-156. The authors mention the “outstanding mechanical properties” of the 3D printed objects without making any comparisons to other 3D printing material (such as PLA) or to materials the 3D object may replace.

Line 161: The authors refer to Supplementary Figure 7 when discussing the role/effect of NaCl.

However, based on Fig. S7, while adding DI water will dissolve NaCl, it is unclear where the Cl⁻ ions end up or what role Cl⁻ is playing by examining Figure S7. Please clarify.

Line 162: In Fig S9, it is unclear how the FTIR data shows that metal chelate bridges are in evidence. Please clarify or expand the discussion.

Line 190: The authors humidify the 3D printed object to improve the characteristics of the object. Why is an extra humidification step needed as the reaction originally takes place in water? What prevents the complete reaction in the original water solution?

Line 197-198: Understanding that space is at a premium, it would help the reader to have a comparison or two of the mechanical strengths of the printed object to “objects printed by other 3D printing techniques”

General Editing: rephrase the sentence on lines 144-147, with focus on the “and greater the number of coordination bonds, higher is the thickness...” Possibly Change the word “higher” to “greater”.

Reviewer #2 (Remarks to the Author):

There are a number of aspects of this paper in terms of both novelty and relevance that are less than convincing.

Binder Jet printing is not new. The authors seems to be suggesting that the particular approach to binder jet printing described in the manuscript is completely new but this is not the case. More evidence needs to be presented to support and distinguish the novelty of the work by comparing it to reports from similar processes. At the moment the novelty is not at all clear.

The results presented in the manuscript have limited potential impact to industry because no metal parts could ever be used as printed using this technique (The strength is determined by the binder that weakly connects the metal powder) – sure this approach is ok for creating interesting metal shapes that might sit on an office shelf but the mechanical properties that have been reported - are not surprisingly -

very low. Even the sintered properties are not good. There are many reported studies on alternative metal '3D printing processes' that produce parts with properties that far exceed the properties presented in this paper. There are also gaps in the description of the results - for example – What is the alloy associated with the plots in Fig 4B?

Reviewer #3 (Remarks to the Author):

Comments to Authors:

Overall, the work provides interesting and original results towards growing the body of knowledge in the area of binder development for binder jetting additive manufacturing. There are significant concerns related to the manuscript, which are prohibitive in accepting the manuscript for publication in its present shape.

1. Abstract: In the abstract, the authors state that environmental issues stand to be addressed before commercialization of additive manufacturing is possible. Although this statement leads well into their motivation for the work, the statement in itself is not substantiated, not necessarily true.
2. Introduction: page 2, line 40 is cut-off; cannot assess statement.
3. Introduction: the use of “domestic” printers is often confusing (consumer level? hobbyist?)
4. Introduction: the second paragraph implies that the challenge in technology adoption of metal binder jetting are the binders, with statements that the binder jetting can be adopted for consumer desktop printers. This is not correct, as for binder jetting of metals, the challenge of powder material handling (loading, unloading, sieving, de-powdering) and heat treatments (de-binding and sintering) in environmentally controlled heat treatment facilities, and distortion control and compensation, are the dominating challenges for tech adoption. This is in conflict with the statement that “non-professionals” will have an easier time with this class of tech if non-hazardous binders were available. The need for eco-friendly binders is valid in itself, but it was not appropriately contextualized in terms of motivation.
5. Introduction: The use of non-toxic binders seems to be the driver behind the motivation in this work. The authors have not presented similar works in literature; such works on non-hazardous binders for binder jetting exist. Specifically when canvassing the literature for biomedical applications, where binders were developed for ceramic powders in binder jetting. Such reviews will need to be expanded. Overall, the introduction was not strong in this regard, with lack of proper depth in framing of present research.
6. Results – Eco-friendly BJM3DP: page 3, paragraph 82-84 “Here, the nozzle clogging issue was prevented by avoiding use of a generally preferred method of jetting a binding agent solution, which precipitates inside the inkjet cartridge and clogs the printing nozzle.” This statement is not well-formulated; what do the authors mean? It is common practice to jet a binding agent via binder jetting. In addition, when the metal powder is mixed with another polymer for assisting in the binding process, this is referred to as a solid-phase binding agent. This approach is commonly used in binder jetting for the biomedical field as well, with water-based binders.
7. Results – Eco-friendly BJM3DP: To the best of the reviewer’s knowledge, there are serious concerns in deploying of DI water onto Al pre-cursor powders, as water is reactive with Al, resulting in hydrogen. There are significant concerns related to safety of the material system described herein. The authors

have used Al powder with DI water, resulting in a hazardous process, where the reaction results in hydrogen gas, which the authors have not mentioned nor discussed mitigation. In addition, the Al powder has a ST3 explosion classification rating, where, should ignition occur, this powder may result in a serious explosion hazard, even in small quantities. The combination of an exothermic reaction between Al+DI water, along with the ST3 classification, is of concern. The use of this material system is highly dangerous for any industrial setting, let alone for domestic use. In addition, the reviewers claim that the chelation process is responsible for the Al-O formation, whereas Al powders readily react with water to form Al₂O₃, even in the absence of such agents.

8. Results – Eco-friendly BJM3DP: The authors need to distinguish mechanical testing on “green parts” (before heat treatments) and sintered parts. In addition, the authors claim “outstanding mechanical properties” (page 5, line 154-156) for their green parts; however, the values can be comparable to other green part strength values in literature.

9. Results – various objects 3D printed using NACit chelator: The authors report the use of “coarse Al powder (<355 μm) was used and the minimum thickness of the 3D-printed object was about 1 mm”, there is serious concern that the results presented herein are not industrially relevant, as the powder size distribution for binder jetting is generally in the range of 10-106 μm (general D10-D90), with significant challenges in terms of processability (powder layer consistency, liquid-powder imbibition issues in terms of lateral versus vertical imbibition, stair-step effect coarseness, layer shifting due to improper layer drying, etc.) for layer thicknesses >200 μm. The authors state that evaporation is used to control the binder “bleed”, but it is expected that to reach a 1mm layer thickness, the liquid (due to capillary forces), will spread laterally significantly. The reviewer has a concern that the de-powdering was excessively done to diminish the effect of the “bleed”, and to give the parts a smoother aspect; this would not be appropriate in the context of presenting the resolution and capabilities of the printed objects.

10. Results – various objects 3D printed using NACit chelator: Figure 3d is used to demonstrate that the part has good green part density. A measure of green part density should be done by Archimedes method, by x-ray computed tomography, etc.

11. Results - Post-treatment process and several 3D-printable metals: the heat treatment (de-binding and sintering) was not provided or apparent in the paper, nor in the supplementary data. As such, it is difficult to assess the results. In addition, the authors claim solid phase sintering; again, this is difficult to assess, due to lack of details on the temperature range, the powder size distribution, etc.

12. Results - Post-treatment process and several 3D-printable metals: the methodology for bulk density (bulk porosity) estimation was not provided or apparent in the paper, nor in the supplementary data. As such, it is difficult to assess the results.

13. Results - Post-treatment process and several 3D-printable metals: the statement on page 6, line 205-208 “3D-printed using copper (Cu), iron (Fe), and a titanium–aluminum–vanadium alloy (Ti–6Al–4V). The successful 3D printing using various metals demonstrates that our developed 3D printing system is universally applicable to a wide variety of elemental metals and their alloys.” Is not substantiated with data in this manuscript; furthermore, figure 4c should be moved to a supplementary information section, if the authors wish to retain the figure as part of the manuscript.

14. Discussion: the discussion section is drastically missing, specifically in linking the results to the available literature, to better explain the results at hand. The discussion should describe for instance:

- a. An overview of literature for binder jetting on water-based binders and metals. Describe the types of metals and liquid binders are used and how the research compares to the outcomes presented herein.
- b. Insights into liquid-powder imbibition for water-based binders, and the explanations for expected liquid imbibition phenomena and how it may relate to layer spread quality (layer shifting, bleeding, etc.)
- c. Explanations on how the green part density compares to other green part densities in binder jetting literature
- d. Explanations on what happens to the binding agents during the de-binding and sintering process in terms of thermal decomposition. How does the thermal decomposition compare to other binders in literature? How does the thermal decomposition agents impact the sintering process?

15. Methods: The methodologies were not described in detail, with incomplete descriptions (for instance heat treatment).

Point-by-point responses to the reviewer's comments are listed below:

Reviewer #1 (Remarks to the Author):

The manuscript “A General Fruit Acid Chelation Route for Eco-Friendly and Ambient 3D printing of metals” presents a technique to 3D print metals such as Al, Cu, & Fe while avoiding hazardous binding agents and possibly providing a route to commercializing 3D metal printing. The manuscript will be of interest to a technical as well as academic audience. Taken as a whole, the manuscript is well written and easy to read.

A few questions that need to be considered include:

Lines 154-156. The authors mention the “outstanding mechanical properties” of the 3D printed objects without making any comparisons to other 3D printing material (such as PLA) or to materials the 3D object may replace.

Reply: Thank you for the valuable comment. We used the term “outstanding mechanical properties” to compare the 3D printed object based on the NaCit binder with other binders used in our study. We believe that expressions were not well formulated and as a result it did mislead the reviewers. Therefore, we corrected the corresponding part in the revised manuscript as follows (highlighted in yellow):

Page 5, Lines 166-167: “The improved mechanical properties of 3D-printed object based on NaCit than other chelators in our study can be ...”

We also agree with the reviewer's suggestion that comparisons to other 3D printing materials would be valuable to our study. Additional information for the comparison between our experimental results and other 3D printing materials (**Figures R1 and R2**) is added in the revised manuscript as **Figure 4c** and the **Supplementary Fig. 18**, and discussed in detail in the discussion section of the revised manuscript.

Figure R1. The recent technological progress in BJM3DP techniques, with respect to the mechanical strengths of printed green bodies based on various binder materials¹⁻⁶.

References:

1. Zhou, Z., Lennon, A., Buchanan, F., McCarthy, H. O. & Dunne, N. Binder jetting additive manufacturing of hydroxyapatite powders: Effects of adhesives on geometrical accuracy and green compressive strength. *Addit. Manuf.* **36**, 101645 (2020).

2. Chai, W. et al. The printability of three water based polymeric binders and their effects on the properties of 3D printed hydroxyapatite bone scaffold. *Ceram. Int.* **46**, 6663-6671 (2020).
3. Kunchala, P. & Kappagantula, K. 3D printing high density ceramics using binder jetting with nanoparticle densifiers. *Mater. Des.* **155**, 443-450 (2018).
4. Tang, Y., Huang, Z., Yang, J. & Xie, Y. Enhancing the Capillary Force of Binder-Jetting Printing Ti6Al4V and Mechanical Properties under High Temperature Sintering by Mixing Fine Powder. *Metals* **10**, 1354 (2020).
5. Solis, D. M., Silva, A. V., Volpato, N. & Berti, L. F. Reaction-bonding of aluminum oxide processed by binder jetting. *J. Manuf. Process.* **41**, 267-272 (2019).
6. Gilmer, D. et al. An in-situ crosslinking binder for binder jet additive manufacturing. *Addit. Manuf.* **35**, 101341 (2020).

Figure R2. The recent technological progress in metal 3D printing techniques with respect to the mechanical strengths of objects⁷⁻¹¹. Each study is based on different techniques as well as different metal powder such as Ti-6Al-4V and NiTi.

References:

7. Yan, C. et al. Evaluation of light-weight AlSi10Mg periodic cellular lattice structures fabricated via direct metal laser sintering. *J. Mater. Process. Technol.* **214**, 856-864 (2014).
8. Challis, V. J. et al. High specific strength and stiffness structures produced using selective laser melting. *Mater. Des.* **63**, 783-788 (2014).
9. Speirs, M., Van Hooreweder, B., Van Humbeeck, J. & Kruth, J. P. Fatigue behaviour of NiTi shape memory alloy scaffolds produced by SLM, a unit cell design comparison. *J. Mech. Behav. Biomed.* **70**, 53-59 (2017).
10. Prashanth, K. G. et al. Production of high strength Al₈₅Nd₈Ni₅Co₂ alloy by selective laser melting. *Addit. Manuf.* **6**, 1-5 (2015).
11. Wiria, F. E. et al. Printing of Titanium implant prototype. *Mater. Des.* **31**, S101-S105 (2010).

Line 161: The authors refer to Supplementary Fig. 7 when discussing the role/effect of NaCl. However, based on Fig. S7, while adding DI water will dissolve NaCl, it is unclear where the Cl⁻ ions end up or what role Cl⁻ is playing by examining Figure S7. Please clarify.

Reply: Thank you for the insightful comment. We agree that a more detailed description of the NaCl chelating mechanism is necessary to clarify our argument for the addition of NaCl additive.

The **Supplementary Fig. 7** in the original manuscript was intended to illustrate how the addition of NaCl can boost the chelation of NaCit with Al surfaces. NaCl dissolved in the water firstly infiltrates through the gap of the naturally existing Al₂O₃ layer, and then react with the aluminum layer to generate AlCl₃. As a result, the reaction enlarges the surface area of aluminum particles and therefore makes them more vulnerable chelation reaction sites. After the process is completed, Cl⁻ ions mostly end up back in the form of NaCl with a trace amount of AlCl₃ left in the system, as can be seen in the XPS spectra in **Figure R3**. The addition of NaCl alone, on the other hand, did not provide any binding properties so that the physical formulation of the 3D object was not possible. We included data in **Figure R3** in the **Supplementary Fig. 7b**, and the relevant information about the fate of Cl⁻ ions as follows (highlighted in yellow):

Supplementary Fig. 7: “a Schematic illustration of metal chelation in presence of NaCl additive. b XPS spectra for Na 1s and Cl 2p of the printed Al object. After the process is completed, Cl⁻ ions mostly end up back in the form of NaCl with a trace amount of AlCl₃ left in the system.”

Figure R3. Na 1s and Cl 2p XPS spectra of 3D printed object using NaCit and NaCl additive.

Line 162: In Fig S9, it is unclear how the FTIR data shows that metal chelate bridges are in evidence. Please clarify or expand the discussion.

Reply: Thank you for the comment. The rationale behind the interpretation of FT-IR spectra was described on Lines 118-137 (page 4) of the original manuscript, which was based on the works by Papageorgiou et al., (reference 47) and by Nakamoto et al. (reference 48). As the reviewer suggested, we also have included a detailed explanation in the revised supplementary information as follows (highlighted in yellow):

Supplementary Fig. 9: “...and NaCl additive. In the FT-IR spectra, the vibrational frequency difference ($\Delta\nu$) between asymmetric (ν_{as}) and symmetric (ν_{s}) stretching vibrations of COO⁻ reflects the change of molecular mass upon the reaction of aluminum and COO⁻, and therefore the change in $\Delta\nu$ for different chelating reactions can represent the degree of chelation. With NaCl additive, the $\Delta\nu$ was decreased from 198 cm⁻¹ (only NaCit) to 172 cm⁻¹, which is a clear sign of increased degree of chelation. ν and $\Delta\nu$ values for different systems are summarized in the Supplementary Table 1.”

Line 190: *The authors humidify the 3D printed object to improve the characteristics of the object. Why is an extra humidification step needed as the reaction originally takes place in water? What prevents the complete reaction in the original water solution?*

Reply: Thank you for the insightful comment. As the reviewer mentioned, we agree that the necessity of the extra humidification step to further enhance the strength of the printed object needs to be justified.

Besides securing strengths of the object, another important criterion in BJM3DPs is minimizing the “Bleeding effect”, which is the result of adverse spreading of the jetted binder solution or water. Increments in the “Bleeding” inevitably ends up in diminishing the resolution of the whole printing process. In order to avoid this phenomenon, we chose to use a marginal amount of water in droplets (33 pL) in an elevated evaporation condition (70 °C) during the initial printing process. After the first round of the chelation, we then applied the humidification step to complete the reaction between the unreacted surplus binders and metal particles, and thus we could achieve both strengths and resolutions through our approach. We included a further demonstration of the “Bleeding Effect” in the revised supplementary information (**Supplementary Fig. 14**). We also included the justification of humidification step in the revised manuscript as follows (highlighted in yellow):

Page 7, Line 223-227: “...SEM image of the humidified 3D-printed object). Since we chose to use a marginal water droplet size (33 pL) at elevated evaporation conditions (builder/nozzle temperature of 70 °C) to minimize the bleeding effect (see **Supplementary Fig. 14** for details), an additional step of the humidification can complete the reaction between surplus chelator and particles, and indeed boosts the strengths of the printed objects without sacrificing the printing resolution. Subsequent thermal...”

Line 197-198: *Understanding that space is at a premium, it would help the reader to have a comparison or two of the mechanical strengths of the printed object to “objects printed by other 3D printing techniques”*

Reply: Thank you for the valuable comment. We agree that a comparison among various 3D printing techniques would provide readers a broader perspective and a better understanding of our work. We have included recent progresses in BJM3DPs (**Figure R1**) as well as other 3D printing techniques (**Figure R2**) in the revised supplementary information (**Figure 4c** and **Supplementary Fig. 18**), and discussed in detail in the discussion section of the revised manuscript.

On the other hand, although the aim of our work is to demonstrate the possibility of a more eco-friendly binder material in metal BJM3DPs and the corresponding ambient chelation mechanism, we came to believe that a better demonstration of feasible property enhancements in the actual printed object would also be helpful in justifying our approach. Therefore, we further optimized the printing condition with smaller and well-packed Al particles (a mixture of particles with average sizes of 10 μm and 75 μm, shown in **Figure R4a** to **R4c**, respectively) to more clearly demonstrate the effectiveness of our approach. Since the mixing enables a better packing density of the metal particles (**Figure R5**), the optimization enabled higher compressive strengths comparable or higher to/than Al objects printed by other techniques (**Figure R4d** and **R4e**). The enhancement in the packing density (lowered void density) were also observed through the optimization (**Figure R4f**). We therefore included the additional information in the revised manuscript as the **Supplementary Fig. 11, 15, 16** and **17**. We further added descriptions in the revised manuscript as follows (highlighted in yellow):

Pages 6-7, Lines 204-216: “...., thereby providing structural integrity. On the other hand, to illustrate prospects of further enhancing the mechanical strengths of BJM3DP objects, we optimized the distribution of the powder sizes and compositions^{55,56}. In regards to overcoming the low intrinsic porosity of BJM3DP objects (30-60 % in general) and reducing microscopic void ratios, we used a bimodal powder distribution based on 10 μm and 75 μm metal particles (**Supplementary Fig. 16**). The improved packing of the metal powders from the filling of interstitial voids with fine particles (**Supplementary Fig. 15** and **17**) thus led void ratios to be critically reduced (1.55 % and 0.30 % for green and sintered body, respectively). As a result, greatly improved mechanical strengths and compressive modulus of green body up to 6.06 MPa and 218.43 MPa, respectively, were obtained. After completing thermal debinding and sintering processes for printed green bodies (**Figure**

4a), the values could further be improved to 29.59 MPa and 1.49 GPa, respectively (**Figure 4b**). These results are indeed much superior than the values acquired in previous reports on green-solvent-soluble binders, and are even comparable to the Al (or Al alloy)-based metal objects printed by other much more sophisticated 3DP methods based on high powered lasers (**Figure 4c** and **Supplementary Fig. 18**)”

Figure R4. Powder size distribution of **a** 10 µm, **b** 75 µm and **c** 355 µm powders. The ISO 13320 laser diffraction method (Beckman Coulter LS-13-320) was used for the analysis. Mechanical strengths of **d** green bodies and **e** sintered bodies based on Al particles with different size distributions, as well as **f** the void ratio extracted from the 3D µCT imaging. An optimized NaCit/NaCl-based chelation route is adopted for all cases.

Figure R5. SEM images of metal particles, green body and sintered body used for further study.

General Editing: rephrase the sentence on lines 144-147, with focus on the “and greater the number of coordination bonds, higher is the thickness...” Possibly Change the word “higher” to “greater”.

Reply: As suggested, we have replaced the word “higher” with “greater” in the mentioned sentence.

Reviewer #2 (Remarks to the Author):

There are a number of aspects of this paper in terms of both novelty and relevance that are less than convincing.

Binder Jet printing is not new. The authors seems to be suggesting that the particular approach to binder jet printing described in the manuscript is completely new but this is not the case. More evidence needs to be presented to support and distinguish the novelty of the work by comparing it to reports from similar processes. At the moment the novelty is not at all clear.

The results presented in the manuscript have limited potential impact to industry because no metal parts could ever be used as printed using this technique (The strength is determined by the binder that weakly connects the metal powder) – sure this approach is ok for creating interesting metal shapes that might sit on an office shelf but the mechanical properties that have been reported - are not surprisingly - very low.

Even the sintered properties are not good. There are many reported studies on alternative metal '3D printing processes' that produce parts with properties that far exceed the properties presented in this paper.

There are also gaps in the description of the results - for example – What is the alloy associated with the plots in Fig 4B?

Reply: We appreciate the reviewer's comments on the relevance of this work in the field of metal 3DP, which indeed helped us to put this work into perspective. As the reviewer commented, BJM3DP is a not a new technique, and there are other metal 3DP techniques that have been developed to give better mechanical properties feasible even for industrial applications. However, it should also be acknowledged that each technological branch has its own academic merits and possesses potentials for a technological breakthrough, even though not all of them provide the best performance right now. BJM3DP is, we believe, one of the most approachable platforms for operationally low-cost, simple and safe metal 3DP processes than other advanced and sophisticated techniques such as the direct metal laser sintering and the selective laser sintering. Moreover, as a groundwork for BJM3DP to become a green technology adequate for both industrial and personalized uses such as rapid prototyping befitting the current 4th industrial revolution, one of the important aspects to be addressed is adapting and understanding of eco-friendly ingredients including binders. In this perspective, we demonstrate the possibility of a greener binder material in metal BJM3DPs and the corresponding ambient chelation mechanism. In specific, a first demonstration of a common citric acid as a solid-phase chelating binder for metals, which can be processed through a simple mixing followed by a 3D structuring with pure water jetting, is presented in our study, and a corresponding water-based chelating mechanism is investigated for a broader applicability.

Furthermore, even though the novelty of our work does not lie in demonstrating the best mechanical properties of the 3D printed metal objects, we came to believe that a better exhibition of feasible property enhancements in the actual printed object would also be helpful in justifying our approach. By choosing metal seed particles with better size distributions and combinations (**Figure R1** below), we were able to indeed increase the strength of 3D printed aluminum objects above 30 MPa in compressive strengths with a marginal void ratio below 0.3% (acquired by 3D μ CT, **Figure R2**). Considering that the pure aluminum (1100~1080A grade, >99% purity) metal bodies commonly exhibit mechanical strengths in a range of 30 to 150 MPa [ASM Handbook Committee, *ASM Handbook, Volume 2: Properties and Selection: Nonferrous Alloys and Special-Purpose Materials*, p 62-122 (ASM International, 1990).], our result is not only better than other reported BJM3DP results (**Figure R3**), but also has shown that it is possible to reach a reasonable value for use in various common applications of aluminum objects. It is true that other works reported mechanical strengths reaching 1000 MPa and void ratio within 0.2~0.5% by using laser-based techniques with tougher metal powder such as titanium alloys (**Figure R4**). However, we believe that our revised results can sufficiently demonstrate the feasibility and relevance of BJM3DPs in the field of metal 3DPs, even in terms of the mechanical properties. We also have demonstrated that our chelation chemistry can be applied to various metals including Cu, Fe and Ti-6Al-4V, not just Al. We included additional data in **Figure R1-R4** in the revised manuscript and the supplementary information (**Figure 4c, Supplementary Fig. 11, 16, and 18**). Furthermore, we revised the

abstract and the introduction to enhance the clarity of our motivations, and to put more emphasis on the advantages of BJM3DPs in the field of metal 3DPs.

Abstract, Lines 25-27: "... the fourth industrial revolution. In this regard, its feasibility of becoming a green technology, which is not an inherent aspect of AM, is gaining more interests. A particular interest in adapting and understanding of eco-friendly ingredients can set its important groundworks. Here, ..."

Page 2, Lines 37-39: "Additive manufacturing (AM), also known as 3D printing, is one of the most important technologies in the fourth industrial revolution because it can enable personalization of products and rapid prototyping. In an attempt to expand the boundaries of AM, numerous researchers have focused on developing printable materials¹⁻³ and corresponding techniques for three-dimensional (3D) printing⁴⁻⁷."

Page 2, Line 49-58: "...which results in bond formation between particles²³⁻²⁵. The technological challenges for the commercial adaptation of BJM3DP still involves overcoming demanding conditions of metal AM processes including materials handling, post-treatments and quality control. However, BJM3DP has particular advantages over other metal 3DPs, stemming from its high feasibility toward operationally low-cost, simple and safe 3DP processes²⁶. The ambient conditions of initial printing process²⁷⁻³⁰ as well as the possible use of commercially available ink cartridges constitute high accessibility to this technology than others, which can potentially facilitate consumer-level desktop applications. Moreover, as a groundwork for BJM3DP to become a more accessible green technology adequate for both industrial and personalized uses such as rapid prototyping, one of the important aspects to be explored is adapting and understanding of environmentally friendly ingredients including binder materials. Two most commonly..."

Page 2, Lines 62-69: "...have an adverse environmental impact^{33,34}. In the field of ceramic BJ3DPs, especially for biomedical applications, there already are various investigations regarding the use of non-hazardous binder materials such as green-solvent-soluble polymers, maltodextrin, sugar and corn starch³⁵⁻³⁷. However, only a few candidates for metal BJ3DP has been explored so far, and the reported characteristics of printed objects such as the porosity and mechanical strengths are far below the ones based on aforementioned common binder materials³⁸. Therefore, it is now imperative to broaden the technological horizon thorough developing new green binding agents for metals that can be eco-friendly as well as non-hazardous^{24,39-41}, with prospects of simultaneously achieving the desired properties of printed objects."

Pages 6-7, Lines 204-216: "..., thereby providing structural integrity. On the other hand, to illustrate prospects of further enhancing the mechanical strengths of BJM3DP objects, we optimized the distribution of the powder sizes and compositions^{57,58}. In regards to overcoming the low intrinsic porosity of BJM3DP objects (30-60 % in general) and reducing microscopic void ratios, we used a bimodal powder distribution based on 10 μm and 75 μm metal particles (**Supplementary Fig. 16**). The improved packing of the metal powders from the filling of interstitial voids with fine particles (**Supplementary Fig. 15 and 17**) thus led void ratios to be critically reduced (1.55 % and 0.30 % for green and sintered body, respectively). As a result, greatly improved mechanical strengths and compressive modulus of green body up to 6.06 MPa and 218.43 MPa, respectively, were obtained. After completing thermal debinding and sintering processes for printed green bodies (**Figure 4a**), the values could further be improved to 29.59 MPa and 1.49 GPa, respectively (**Figure 4b**). These results are indeed much superior than the values acquired in previous reports on green-solvent-soluble binders, and are even comparable to the Al (or Al alloy)-based metal objects printed by other much more sophisticated 3DP methods based on high powered lasers (**Figure 4c and Supplementary Fig. 18**)"

Pages 7, Lines 233-244:"These strategies are expected to contribute to improvement in the mechanical properties of printed objects through minimization of the void ratios and porosity in final products. Figure 4d shows objects 3D-printed using copper (Cu), iron (Fe), and a titanium–aluminum–vanadium alloy (Ti–6Al–4V). The printed objects showed structural integrity that cannot be acquired through jetting of water alone without our solid-phase binder mixed with metal particles (**Supplementary Fig. 22**). Furthermore, FT-IR spectra of all printed metal objects exhibit noticeable shifts in $\Delta\nu(\text{COO}^-)$ (188 cm^{-1} for Cu, 187 cm^{-1} for Fe and 189 cm^{-1} for Ti-6Al-4V) compared to pure NaCit (198 cm^{-1}), which agree with the trend we observed in Al-based printed objects (**Figure 4e**). Changes in ionic compositions between pure and BJM3DP-printed

metals were also observed, wherein Cu^{2+} , Fe^{3+} and Al^{3+} can preferentially chelate with COO^- of citrate ions (**Supplementary Fig. 23**). These results demonstrate that our chelator is feasible for forming chelation bridges between metal particles of other metals, and thus can be applied as a binder for BJM3DPs of various metals and alloys than just Al alone.”

Pages 8-9, Lines 247-286: “An eco-friendly chelator-based binding mechanism for BJM3DP was proposed and the feasibility of achieving an effective 3D printing system was demonstrated in this study. In the field of BJM3DPs, a few candidates for green-solvent-soluble binders have been reported so far, such as polyvinyl alcohol (PVA)^{37,42,57}, polyacryl amide (PAM)⁵⁸, polyvinyl pyrrolidone (PVP)⁵⁹, and maltodextrin³⁷, some of which were adopted from successful demonstrations in the the field of ceramic BJ3DPs for biomedical applications⁵⁸. However, the mechanical properties of the printed metal objects based on them, even with highly tough Ti-6Al-4V alloy, were rather less competitive for broad applications compared to the ones based on common binder materials developed without considerations for eco-friendliness. Among them, dextrin, a liquid-based binder, has shown promises that can provide meaningful mechanical properties to the printed green bodies of Al/alumina (**Figure 4c**). As a solid-phase binder, PVA has been adopted but the mechanical properties achieved was abysmal⁵⁷ contrary to previous demonstrations of reasonable binding properties in ceramic (hydroxyapatite) BJ3DPs³⁷. In comparison, results in this work demonstrated that the citric acid, as an another solid-phase green binder that only requires the jetting of pure water for the structural formation, can endorse prospects of simultaneously achieving desired mechanical properties of printed objects as well as the eco-friendly and facile printing processing. By using citric acids as binders, the structural integrity of the 3D-printed objects can be achieved through efficient formation of metal-chelate bridges between metal particles. A combination of optimized powder size distribution and the efficient binding properties of citric acid, therefore, enabled high packing densities for both green and sintered body of printed Al (1.55 % and 0.30% void ratios, respectively). The optimized packing then led to the achieved good mechanical strengths (~30 MPa in compressive strength) for Al objects, which were comparable to or even higher than the ones printed by other sophisticated metal 3DPs based on high powered lasers (**Supplementary Fig. 18**). The porosity of the green body, on the other hand, still has a room for enhancements, even though the value (41 %) itself is well within the previous reports (30-60 %) ⁶⁰⁻⁶². Furthermore, the resolution of the 3D printed objects, which is another important aspect along with mechanical strengths, can be enhanced by controlling the evaporation of jetted water. The inevitable liquid-powder imbibition could be reduced by elevating the builder temperature near evaporation point (70 °C) of water, of which the plausible insufficiency in the metal bridging from the fast evaporation was compensated by using an additional step of the humification which completes the remnant chelation reactions between metal particles and citrates (**Supplementary Fig. 19**). The whole printing process based on citric-acid-based BJM3DP can be completed through the thermal treatments of green bodies, where the debinding and sintering result in the merging of metal particles and thus provides an abrupt enhancement in the mechanical strengths. In order to minimize the rapid shrinkage and maintain the structural integrity during the initial debinding process, the optimization of debinding temperature and duration is required to remove an appropriate amount of chelation bridges at mild rate. Since the citric acid thermally decomposes above 175 °C, which is far lower than that of other water-soluble polymeric binders such as PVP, it is necessary to choose lower debinding temperature. A fine adjustments and optimizations are out-of-scope for this work and would require further study, but our choice of conditions (350 °C for 3 hours) did not sacrifice the structural integrity. This work also demonstrated that, the chelation chemistry of citric acid observed in Al could be expanded to various metals such as Cu, Fe and Ti-Al6-4V alloy. Based on these aspects, the proposed facile approach of using environmentally friendly chelators is expected be a cornerstone for promoting the development of the highly approachable, low-cost and safe consumer-level desktop metal 3D printing systems.”

Figure R1. Powder size distribution of **a** 10 μm, **b** 75 μm and **c** 355 μm powders. The ISO 13320 laser diffraction method (Beckman Coulter LS-13-320) was used for the analysis.

Figure R2. **a** The void ratio, and mechanical strengths of **b** green bodies and **c** sintered bodies based on Al particles with different size distributions. An optimized NaCit/NaCl-based chelation route is adopted for all cases. The void ratio of printed objects was acquired by using 3D computed tomography (μ CT, Nikon XTH 320) with the voxel size of 1 μm at X-ray beam energy of 210 kV. Voxel analysis were performed by using VGSTUDIO MAX (Volume Graphics Pte. Ltd).

Figure R3. The recent technological progress in BJM3DP techniques, with respect to the mechanical strengths of printed green bodies based on various binder materials¹⁻⁶.

References:

1. Zhou, Z., Lennon, A., Buchanan, F., McCarthy, H. O. & Dunne, N. Binder jetting additive manufacturing of hydroxyapatite powders: Effects of adhesives on geometrical accuracy and green compressive strength. *Addit. Manuf.* **36**, 101645 (2020).
2. Chai, W. et al. The printability of three water based polymeric binders and their effects on the properties of 3D printed hydroxyapatite bone scaffold. *Ceram. Int.* **46**, 6663-6671 (2020).
3. Kunchala, P. & Kappagantula, K. 3D printing high density ceramics using binder jetting with nanoparticle densifiers. *Mater. Des.* **155**, 443-450 (2018).
4. Tang, Y., Huang, Z., Yang, J. & Xie, Y. Enhancing the Capillary Force of Binder-Jetting Printing Ti6Al4V and Mechanical Properties under High Temperature Sintering by Mixing Fine Powder. *Metals* **10**, 1354 (2020).
5. Solis, D. M., Silva, A. V., Volpato, N. & Berti, L. F. Reaction-bonding of aluminum oxide processed by binder jetting. *J. Manuf. Process.* **41**, 267-272 (2019).
6. Gilmer, D. et al. An in-situ crosslinking binder for binder jet additive manufacturing. *Addit. Manuf.* **35**, 101341 (2020).

Figure R4. The recent technological progress in metal 3D printing techniques with respect to the mechanical strengths of objects⁷⁻¹¹. Each study is based on different techniques as well as different metal powder such as Ti-6Al-4V and NiTi.

Reference:

7. Yan, C. et al. Evaluation of light-weight AlSi10Mg periodic cellular lattice structures fabricated via direct metal laser sintering. *J. Mater. Process. Technol.* **214**, 856-864 (2014).
8. Challis, V. J. et al. High specific strength and stiffness structures produced using selective laser melting. *Mater. Des.* **63**, 783-788 (2014).
9. Speirs, M., Van Hooreweder, B., Van Humbeeck, J. & Kruth, J. P. Fatigue behaviour of NiTi shape memory alloy scaffolds produced by SLM, a unit cell design comparison. *J. Mech. Behav. Biomed.* **70**, 53-59 (2017).
10. Prashanth, K. G. et al. Production of high strength Al85Nd8Ni5Co2 alloy by selective laser melting. *Addit. Manuf.* **6**, 1-5 (2015).
11. Wiria, F. E. et al. Printing of Titanium implant prototype. *Mater. Des.* **31**, S101-S105 (2010).

Reviewer #3 (Remarks to the Author):

Overall, the work provides interesting and original results towards growing the body of knowledge in the area of binder development for binder jetting additive manufacturing. There are significant concerns related to the manuscript, which are prohibitive in accepting the manuscript for publication in its present shape.

1. Abstract: In the abstract, the authors state that environmental issues stand to be addressed before commercialization of additive manufacturing is possible. Although this statement leads well into their motivation for the work, the statement in itself is not substantiated, not necessarily true.

Reply: Thank you for sharing your expertise. We agree that the abstract oversimplified the issues around the commercialization of AM. We believe that the technique we incorporated, BJM3DP, has technological advantages that stem from its feasibility in operationally low-cost, simple and safe adaptation of metal 3DPs for commercial applications, which lead to our motivation that developments toward greener processing comprise one of important technological nodes that needs to be looked into. We have toned down the abstract in the revised manuscript as follows:

Abstract, Lines 25-27: “... the fourth industrial revolution. In this regard, its feasibility of becoming a green technology, which is not an inherent aspect of AM, is gaining more interests. A particular interest in adapting and understanding of eco-friendly ingredients can set its important groundworks. Here, we...”

2. Introduction: page 2, line 40 is cut-off; cannot assess statement.

Reply: We apologize for the ill-preparation of the manuscript file. There seems to be an error during the file conversion process. We have corrected the sentence in the revised manuscript as follows:

Page 2, Lines 37-39: “Additive manufacturing (AM), also known as 3D printing, is one of the most important technologies in the fourth industrial revolution because it can enable personalization of products and rapid prototyping. In an attempt to expand the boundaries of AM, numerous researchers have focused on developing printable materials¹⁻³ and corresponding techniques for three-dimensional (3D) printing⁴⁻⁷.”

3. Introduction: the use of “domestic” printers is often confusing (consumer level? hobbyist?)

Reply: Thank you for the comment. We have replaced the term “domestic” with “consumer-level” and “personalized use” on pages 1 (abstract), 2 (introduction) and 9 (discussion).

4. Introduction: the second paragraph implies that the challenge in technology adoption of metal binder jetting are the binders, with statements that the binder jetting can be adopted for consumer desktop printers. This is not correct, as for binder jetting of metals, the challenge of powder material handling (loading, unloading, sieving, de-powdering) and heat treatments (de-binding and sintering) in environmentally controlled heat treatment facilities, and distortion control and compensation, are the dominating challenges for tech adoption. This is in conflict with the statement that “non-professionals” will have an easier time with this class of tech if non-hazardous binders were available. The need for eco-friendly binders is valid I itself, but it was not appropriately contextualized in terms of motivation.

Reply: Thank you for the insightful comment. We also believe that there is a need for eco-friendly binders in BJM3DPs, which we have improperly contextualized and oversimplified in the original manuscript. We have changed the second paragraph of the introduction in the revised manuscript as follows (highlighted in yellow):

Page 2, Lines 49-58: “...which results in bond formation between particles²³⁻²⁵. The technological challenges for the commercial adaptation of BJM3DP still involves overcoming demanding conditions of metal AM processes including materials handling, post-treatments and quality control. However, BJM3DP has particular advantages over other metal 3DPs, stemming from its high feasibility toward operationally low-cost, simple

and safe 3DP processes²⁶. The ambient conditions of initial printing process²⁷⁻³⁰ as well as the possible use of commercially available ink cartridges constitute high accessibility to this technology than others, which can potentially facilitate consumer-level desktop applications. Moreover, as a groundwork for BJM3DP to become a more accessible green technology adequate for both industrial and personalized uses such as rapid prototyping, one of the important aspects to be explored is adapting and understanding of environmentally friendly ingredients including binder materials. Two most commonly...

5. Introduction: *The use of non-toxic binders seems to be the driver behind the motivation in this work. The authors have not presented similar works in literature; such works on non-hazardous binders for binder jetting exist. Specifically when canvassing the literature for biomedical applications, where binders were developed for ceramic powders in binder jetting. Such reviews will need to be expanded. Overall, the introduction was not strong in this regard, with lack of proper depth in framing of present research.*

Reply: We appreciate your expertise on this matter. It is true that there exist non-hazardous binders largely used for ceramics. Moreover, several papers have also reported the use of non- or less-hazardous binders than commonly used ones for metals too. In this regard, we have reviewed related literatures and added corresponding references/descriptions to the introduction part of the revised manuscript as follows (highlighted in yellow):

Page 2, Lines 62-69: "...have an adverse environmental impact^{33,34}. In the field of ceramic BJ3DPs, especially for biomedical applications, there already are various investigations regarding the use of non-hazardous binder materials such as green-solvent-soluble polymers, maltodextrin, sugar and corn starch³⁵⁻³⁷. However, only a few candidates for metal BJ3DP has been explored so far, and the reported characteristics of printed objects such as the porosity and mechanical strengths are far below the ones based on aforementioned common binder materials³⁸. Therefore, it is now imperative to broaden the technological horizon through developing new green binding agents for metals that can be eco-friendly as well as non-hazardous^{24,39-41}, with prospects of simultaneously achieving the desired properties of printed objects."

6. Results – Eco-friendly BJM3DP: *page 3, paragraph 82-84 "Here, the nozzle clogging issue was prevented by avoiding use of a generally preferred method of jetting a binding agent solution, which precipitates inside the inkjet cartridge and clogs the printing nozzle." This statement is not well-formulated; what do the authors mean? It is common practice to jet a binding agent via binder jetting. In addition, when the metal powder is mixed with another polymer for assisting in the binding process, this is referred to as a solid-phase binding agent. This approach is commonly used in binder jetting for the biomedical field as well, with water-based binders.*

Reply: Thank you for the comment. We reformulated the corresponding sentence in the revised manuscript as follows (highlighted in yellow):

Page 3, Lines 89-92: "...was mixed with the chelator. Here, the nozzle clogging issue from the precipitation of binding agents inside inkjet cartridge was avoided since the chelator functions as a solid-phase binding agent pre-mixed with metal particles, which also is one of commonly adopted approaches in biomedical fields^{35-37,42}. Next, two gantry..."

7. Results – Eco-friendly BJM3DP: *To the best of the reviewer's knowledge, there are serious concerns in deploying of DI water onto Al pre-cursor powders, as water is reactive with Al, resulting in hydrogen. There are significant concerns related to safety of the material system described herein. The authors have used Al powder with DI water, resulting in a hazardous process, where the reaction results in hydrogen gas, which the authors have not mentioned nor discussed mitigation. In addition, the Al powder has a ST3 explosion classification rating, where, should ignition occur, this powder may result in a serious explosion hazard, even*

in small quantities. The combination of an exothermic reaction between Al+DI water, along with the ST3 classification, is of concern. The use of this material system is highly dangerous for any industrial setting, let alone for domestic use. In addition, the reviewers claim that the chelation process is responsible for the Al-O formation, whereas Al powders readily react with water to form Al₂O₃, even in the absence of such agents.

Reply: Thank you for the valuable comment. Indeed, there could very well be safety issues regarding the water reactivity and ST3 explosion classification of Al powder. As the reviewer pointed out, we also believe that potential hazards regarding the Al-based system needs to be discussed. Furthermore, proper equipping of ventilation systems and gas detectors are highly recommended regardless of used metal sources in BJM3DPs. Therefore, we performed additional experiments to gauge the supposed danger in our system.

We firstly examined the generation of hydrogen gas during the process. By using water displacement method, we measured the amount of hydrogen generated through the reaction between water and Al in a closed system, compared it with our real printing system, and then estimated the danger based on the lower explosive limit (LEL) of hydrogen (41,000 ppm). We found that 193 ppm/hour of hydrogen can be generated in our system (250 ml of water jetting over 6 hours in a cubic space of 0.6×0.6×0.6 m³ volume) without ventilation, which would need 200 hours of continuous processing. Even though our system itself generates marginal amount of hydrogen far below LEL, it is true that proper ventilation and equipping of hydrogen detectors are required for safe operations because continuous operation for fabricating complicated object can indeed pose danger regardless of whether it is consumer-level or industry-level applications. On the other hand, since the main point of our work is to introduce a possible candidate for better binder materials for metals, we have to mention that its applicability is not limited to only Al system (**Figure 4** of the manuscript). Choice of different metal would be helpful in reducing the potential explosion/ignition danger. We included a disclaimer regarding the danger of using Al and water in the revised manuscript as follows (highlighted in yellow):

*Page 4, Lines 106-112: "... overall metal 3D printing process are shown in **Supplementary Fig. 2**. As a safety disclaimer, the Al powder is classified with ST3 explosive ratings, which can pose a potential danger for Al powder-based systems. Furthermore, a plausible reaction between water and Al could also generate combustible hydrogen during the process, thus a proper ventilation and an equipment of hydrogen detectors are required for the safe operation. In our experimental system (capable of jetting 250 ml of water over 6 hours), 193 ppm/hour of hydrogen generation was observed, which would indeed pose danger over 200 hours of continuous printing in a completely closed system by reaching lower explosive limit of the hydrogen (41,000 ppm)⁴³."*

As for the Al-O formation through chelation, the natural existence of Al₂O₃ actually interferes with the binding of metal particles (**Figure 2a** of the manuscript), which can easily be seen from the lack of structural formulation upon water jetting without binders (**Figure R1**). Therefore, we needed to use additional Cl⁻ ions to facilitate the reaction between our chelators and metal to make Al-O bridge between Al and NaCit (**Figure 2b** of the manuscript). We added the data in **Figure R1** to the revised supplementary information (**Supplementary Fig. 22**)

Figure R1. Structure formulations of Al powder upon spray D.I. water. **a** Al-NaCit system, **b** Al-only system. After the spraying water on the powder through pattern (YONSEI), the Al-only system does not maintain its pattern when dusted by an air blowgun.

8. Results – Eco-friendly BJM3DP: The authors need to distinguish mechanical testing on “green parts” (before heat treatments) and sintered parts. In addition, the authors claim “outstanding mechanical properties” (page 5, line 154-156) for their green parts; however, the values can be comparable to other green part strength values in literature.

Reply: Thank you for the detailed comment. We have revised the manuscript accordingly to the reviewer’s comment by distinguishing green parts and sintered parts in the revised manuscript.

For the second part of the comment, we used the term “outstanding” to compare the mechanical properties among the 4 chelators used in our system. We revised the corresponding sentence to avoid confusion as follows (highlighted in yellow):

Page 5, Lines 166-167: “The improved mechanical properties of 3D-printed object based on NaCit than other chelators in our study can be ...”

Moreover, we further incorporated a comparison between our system and the literature (**Figure R2** and **R3**) in the revised manuscript, to gauge the potential of our work in a broader perspective. We have included detailed discussions in the discussion section of the revised manuscript.

Figure R2. The recent technological progress in BJM3DP techniques, with respect to the mechanical strengths of printed green bodies based on various binder materials¹⁻⁶.

Reference:

1. Zhou, Z., Lennon, A., Buchanan, F., McCarthy, H. O. & Dunne, N. Binder jetting additive manufacturing of hydroxyapatite powders: Effects of adhesives on geometrical accuracy and green compressive strength. *Addit. Manuf.* **36**, 101645 (2020).
2. Chai, W. et al. The printability of three water based polymeric binders and their effects on the properties of 3D printed hydroxyapatite bone scaffold. *Ceram. Int.* **46**, 6663-6671 (2020).
3. Kunchala, P. & Kappagantula, K. 3D printing high density ceramics using binder jetting with nanoparticle densifiers. *Mater. Des.* **155**, 443-450 (2018).
4. Tang, Y., Huang, Z., Yang, J. & Xie, Y. Enhancing the Capillary Force of Binder-Jetting Printing Ti6Al4V

and Mechanical Properties under High Temperature Sintering by Mixing Fine Powder. *Metals* **10**, 1354 (2020).

5. Solis, D. M., Silva, A. V., Volpato, N. & Berti, L. F. Reaction-bonding of aluminum oxide processed by binder jetting. *J. Manuf. Process.* **41**, 267-272 (2019).
6. Gilmer, D. et al. An in-situ crosslinking binder for binder jet additive manufacturing. *Addit. Manuf.* **35**, 101341 (2020).

Figure R3. The recent technological progress in metal 3D printing techniques with respect to the mechanical strengths of objects⁷⁻¹¹. Each study is based on different techniques as well as different metal powder such as Ti-6Al-4V and NiTi.

Reference:

7. Yan, C. et al. Evaluation of light-weight AlSi10Mg periodic cellular lattice structures fabricated via direct metal laser sintering. *J. Mater. Process. Technol.* **214**, 856-864 (2014).
8. Challis, V. J. et al. High specific strength and stiffness structures produced using selective laser melting. *Mater. Des.* **63**, 783-788 (2014).
9. Speirs, M., Van Hooreweder, B., Van Humbeeck, J. & Kruth, J. P. Fatigue behaviour of NiTi shape memory alloy scaffolds produced by SLM, a unit cell design comparison. *J. Mech. Behav. Biomed.* **70**, 53-59 (2017).
10. Prashanth, K. G. et al. Production of high strength Al85Nd8Ni5Co2 alloy by selective laser melting. *Addit. Manuf.* **6**, 1-5 (2015).
11. Wiria, F. E. et al. Printing of Titanium implant prototype. *Mater. Des.* **31**, S101-S105 (2010).

9. Results – various objects 3D printed using NACit chelator: The authors report the use of “coarse Al powder (<355 μm) was used and the minimum thickness of the 3D-printed object was about 1 mm”, there is serious concern that the results presented herein are not industrially relevant, as the powder size distribution for binder jetting is generally in the range of 10-106 μm (general D10-D90), with significant challenges in terms of processability (powder layer consistency, liquid-powder imbibition issues in terms of lateral versus vertical imbibition, stair-step effect coarseness, layer shifting due to improper layer drying, etc.) for layer thicknesses >200 μm. The authors state that evaporation is used to control the binder “bleed”, but it is expected that to reach a 1mm layer thickness, the liquid (due to capillary forces), will spread laterally significantly. The

reviewer has a concern that the de-powdering was excessively done to diminish the effect of the “bleed”, and to give the parts a smoother aspect; this would not be appropriate in the context of presenting the resolution and capabilities of the printed objects.

Reply: We appreciate the reviewer for sharing expertise on the processing challenges in improving the resolution of 3D printed objects. We have included additional experiments on “Bleeding Effect” depending on the printing conditions in the revised supplementary information (**Supplementary Fig. 14**). Elevating the printing temperature enables the control of the evaporation rate of the water during the process. By doing so, we were able to reduce the vertical and lateral bleeding and to improve the resolution as shown in **Figure R4**. We added the corresponding description to the manuscript as follows (highlighted in yellow):

Page 7, Line 223-227: “... of the humidified 3D-printed object). Since we chose to use a marginal water droplet size (33 pL) at elevated evaporation conditions (builder/nozzle temperature of 70 °C) to minimize the bleeding effect (see **Supplementary Fig. 14** for details), an additional step of the humidification can complete the reaction between surplus chelator and particles, and indeed boosts the strengths of the printed objects without sacrificing the printing resolution. Subsequent thermal sintering ...”

Figure R4. Demonstration of the minimization process for bleeding effects by using different initial printing conditions. The temperature of builder was gradually increased from 10 °C to 80 °C. The scale bar on the image is 10 mm.

Furthermore, regarding the industrial relevance of our results using coarse grained (< 355 μm) Al powder, we conducted additional experiments with finer grain Al powders and further optimized the process. **Figure R5** shows the size distribution of the 10 μm and 75 μm powders (10.25 μm and 78.97 μm on D10 distribution parameter) used for the further study. By using bimodal particles (a mixture of 10 μm and 75 μm powders), we could indeed further optimize the printing process and achieved far improved void ratio (4.51 % to 0.30 % acquired by 3D μCT) and mechanical strength (6.3 MPa to 29.6 MPa in compressive strength), as shown in **Figure R6**. A better packing density in the bimodal particles, we believe, enabled a better packing of metal particles (**Figure R7**) and thus led to lower voids of the final product. The data in **Figure R5** and **R6** are included in the revised supplementary information (**Supplementary Fig. 11 and 16**). We further included related description to the revised manuscript as follows (highlighted in yellow):

Pages 6-7, Lines 204-216: “..., thereby providing structural integrity. On the other hand, to illustrate prospects of further enhancing the mechanical strengths of BJM3DP objects, we optimized the distribution of the powder sizes and compositions^{55,56}. In regards to overcoming the low intrinsic porosity of BJM3DP objects (30-60 %

in general) and reducing microscopic void ratios, we used a bimodal powder distribution based on 10 μm and 75 μm metal particles (**Supplementary Fig. 16**). The improved packing of the metal powders from the filling of interstitial voids with fine particles (**Supplementary Fig. 15 and 17**) thus led void ratios to be critically reduced (1.55 % and 0.30 % for green and sintered body, respectively). As a result, greatly improved mechanical strengths and compressive modulus of green body up to 6.06 MPa and 218.43 MPa, respectively, were obtained. After completing thermal debinding and sintering processes for printed green bodies (**Figure 4a**), the values could further be improved to 29.59 MPa and 1.49 GPa, respectively (**Figure 4b**). These results are indeed much superior than the values acquired in previous reports on green-solvent-soluble binders, and are even comparable to the Al (or Al alloy)-based metal objects printed by other much more sophisticated 3DP methods based on high powered lasers (**Figure 4c and Supplementary Fig. 18**)”

Figure R5. Powder size distribution of **a** 10 μm and **b** 75 μm powders. The ISO 13320 laser diffraction method (Beckman Coulter LS-13-320) was used for the analysis.

Figure R6. **a** The void ratio and **b** the mechanical strengths of printed objects depending on the size distribution of the metal powder.

Figure R7. SEM images of metal particles, green body and sintered body used for further study, depending on the size and composition distributions of the particles.

10. Results – various objects 3D printed using NACit chelator: Figure 3d is used to demonstrate that the part has good green part density. A measure of green part density should be done by Archimedes method, by x-ray computed tomography, etc.

Reply: Thank you for the comment. We further conducted μ CT imaging to corroborate our argument on the improved green part density (Figure R6 and R8). We have included the data in the revised manuscript and the supplementary information (Supplementary Fig. 11 and 15)

Page 6, Line 199-202: “...as shown in Figure 3c can be acquired. Figure 3d and Supplementary Fig. 15 show the rendered images and extracted void ratios of the printed object from computed 3D X-ray microtomography (3D μ CT) analyses. A good packing density (void ratio of 4.92 %) of the green body based on coarse Al powder (<355 μ m) by itself not only ...”

Figure R8. Rendered images and void ratios from μ CT analyses for **a** green body and **b** sintered body based on 355 μ m Al powders. **c** Rendered images and void ratios from μ CT analyses for green and sintered bodies based on metal particles with various size distributions.

11. Results - Post-treatment process and several 3D-printable metals: the heat treatment (de-binding and sintering) was not provided or apparent in the paper, nor in the supplementary data. As such, it is difficult to assess the results. In addition, the authors claim solid phase sintering; again, this is difficult to assess, due to lack of details on the temperature range, the powder size distribution, etc.

Reply: We have included the heat treatment condition (**Figure R9a**) and the size distribution of metal particles (**Figure R5 and R9b**) in the revised supplementary information as **Supplementary Fig. 20** and **16**. In specific, we performed 3 cycles of the following heat treatment profile: we firstly elevated temperature by 10 $^{\circ}$ C/min to 350 $^{\circ}$ C and maintained the temperature for 3 hours for the debinding. After that, we elevated temperature again by 10 $^{\circ}$ C/min to 620 $^{\circ}$ C and maintained the temperature for 15 hours for the sintering. We further revised the experimental sections as follows (highlighted in yellow):

Supplementary Information, Line 51-55: “...under vacuum (0.03 torr). The heat treatment profiles are shown in **Supplementary Fig. 20**. 3 cycles of the following heat treatment profile are conducted: (Debinding) Heating from RT to 350 $^{\circ}$ C by 10 $^{\circ}$ C/min followed by 350 $^{\circ}$ C treatment for 3 hours. (Sintering) Heating from 350 $^{\circ}$ C to 620 $^{\circ}$ C by 10 $^{\circ}$ C/min followed by 620 $^{\circ}$ C treatment for 15 hours. (Cooling) Cooling from 620 $^{\circ}$ C to RT by -10 $^{\circ}$ C/min.”

Figure R9. **a** The heat treatment profile. **b** The size distribution of the metal particle used in the original manuscript.

12. Results - Post-treatment process and several 3D-printable metals: the methodology for bulk density (bulk porosity) estimation was not provided or apparent in the paper, nor in the supplementary data. As such, it is difficult to assess the results.

Reply: Thank you for the comment. In the original manuscript, we measured the bulk porosity by using the porosimetry with mercury (Quantachrome, PM33GT), which might not reflect the density of possible closed pores. In order to enhance the accuracy of the assessment, we have conducted additional experiments for the measurement of void ratio by using 3D computed tomography (μ CT, Nikon XTH 320), which are shown in **Figure R8**. We added corresponding experimental details in the revised supplementary information as follows (highlighted in yellow):

Supplementary Information, Line 61-64: “...with a compression rate of 8 mm s^{-1} . The packing density of printed objects were acquired by using 3D computed tomography (μ CT, Nikon XTH 320) with the voxel size of $1 \text{ }\mu\text{m}$ at X-ray beam energy of 210 kV. Voxel analysis and 3D visualization were performed by using VGSTUDIO MAX (Volume Graphics Pte. Ltd).”

Furthermore, data and analyses related to the packing density measurement are included in the revised manuscript and the supplementary information (**Figure 3d** and **Supplementary Fig. 15**).

Page 6, Line 199-202: “...as shown in **Figure 3c** can be acquired. **Figure 3d** and **Supplementary Fig. 15** show the rendered images and extracted void ratios of the printed object from computed 3D X-ray microtomography (3D μ CT) analyses. A good packing density (void ratio of 4.92 %) of the green body based on coarse Al powder ($<355 \text{ }\mu\text{m}$) by itself not only ...”

13. Results - Post-treatment process and several 3D-printable metals: the statement on page 6, line 205-208 “3D-printed using copper (Cu), iron (Fe), and a titanium–aluminum–vanadium alloy (Ti–6Al–4V). The successful 3D printing using various metals demonstrates that our developed 3D printing system is universally applicable to a wide variety of elemental metals and their alloys.” Is not substantiated with data in this manuscript; furthermore, figure 4c should be moved to a supplementary information section, if the authors wish to retain the figure as part of the manuscript.

Reply: Thank you for the comment and the suggestion. The **Figure 4c** in the original manuscript was intended to illustrate that our binder material can be applied to various metals to form chelating bridges among particles and thus provide structural integrity to the printed metal objects, as the water jetting without the our solid-phase binder does not provide any structural formations. We performed additional experiments to corroborate the formation of chelation bridges (**Figure R10** and **Figure R11**) and included in the revised manuscript (**Figure 4e**) and supplementary information (**Supplementary Fig. 23**). We also revised the corresponding

sentence in the revised manuscript as follows:

Pages 7, Lines 235-244: "...vanadium alloy (Ti-6Al-4V). The printed objects showed structural integrity that cannot be acquired through jetting of water alone without our solid-phase binder mixed with metal particles (Supplementary Fig. 22). Furthermore, FT-IR spectra of all printed metal objects exhibit noticeable shifts in $\Delta\nu(\text{COO}^-)$ (188 cm^{-1} for Cu, 187 cm^{-1} for Fe and 189 cm^{-1} for Ti-6Al-4V) compared to pure NaCit (198 cm^{-1}), which agrees with the trend we observed in Al-based printed objects (Figure 4e). Changes in ionic compositions between pure and BJM3DP-printed metals were also observed, wherein Cu^{2+} , Fe^{3+} and Al^{3+} can preferentially chelate with COO^- of citrate ions (Supplementary Fig. 23). These results demonstrate that our chelator is feasible for forming chelation bridges between metal particles of other metals, and thus can be applied as a binder for BJM3DPs of various metals and alloys than just Al alone."

Figure R10. The FTIR spectra and corresponding peak separation ($\Delta\nu(\text{COO}^-)$) of 3D printed **a** Cu, **b** Fe and **c** Ti-6Al-4V objects compared to the one for pure NaCit.

Figure R11. The XPS spectra of BJM3DP printed (top) and pure metals (bottom) for **a** Cu, **b** Fe and **c** Ti-6Al-4V.

14. Discussion: the discussion section is drastically missing, specifically in linking the results to the available literature, to better explain the results at hand. The discussion should describe for instance:

a. An overview of literature for binder jetting on water-based binders and metals. Describe the types of metals and liquid binders are used and how the research compares to the outcomes presented herein.

b. Insights into liquid-powder imbibition for water-based binders, and the explanations for expected liquid imbibition phenomena and how it may relate to layer spread quality (layer shifting, bleeding, etc.)

c. Explanations on how the green part density compares to other green part densities in binder jetting literature

d. Explanations on what happens to the binding agents during the de-binding and sintering process in terms of thermal decomposition. How does the thermal decomposition compare to other binders in literature? How does the thermal decomposition agents impact the sintering process?

Reply: Thank you for the in-depth comment. We deeply appreciate your expertise that you shared for us to improve the discussion of our work. As the reviewer suggested, we have rewritten the discussion section in

the revised manuscript to provide a better perspective on the technological advances achieved from this work in the field of BJ3DPs. We also have included literature survey for the recent progress in binder developments and the corresponding physical properties (**Figure 4c** and **Supplementary Fig. 18**).

Discussion

Pages 8-9, Lines 247-286: “An eco-friendly chelator-based binding mechanism for BJM3DP was proposed and the feasibility of achieving an effective 3D printing system was demonstrated in this study. In the field of BJM3DPs, a few candidates for green-solvent-soluble binders have been reported so far, such as polyvinyl alcohol (PVA)^{37,42,57}, polyacryl amide (PAM)⁵⁸, polyvinyl pyrrolidone (PVP)⁵⁹, and maltodextrin³⁷, some of which were adopted from successful demonstrations in the the field of ceramic BJ3DPs for biomedical applications⁵⁸. However, the mechanical properties of the printed metal objects based on them, even with highly tough Ti-6Al-4V alloy, were rather less competitive for broad applications compared to the ones based on common binder materials developed without considerations for eco-friendliness. Among them, dextrin, a liquid-based binder, has shown promises that can provide meaningful mechanical properties to the printed green bodies of Al/alumina (**Figure 4c**). As a solid-phase binder, PVA has been adopted but the mechanical properties achieved was abysmal⁵⁷ contrary to previous demonstrations of reasonable binding properties in ceramic (hydroxyapatite) BJ3DPs³⁷. In comparison, results in this work demonstrated that the citric acid, as an another solid-phase green binder that only requires the jetting of pure water for the structural formation, can endorse prospects of simultaneously achieving desired mechanical properties of printed objects as well as the eco-friendly and facile printing processing. By using citric acids as binders, the structural integrity of the 3D-printed objects can be achieved through efficient formation of metal-chelate bridges between metal particles. A combination of optimized powder size distribution and the efficient binding properties of citric acid, therefore, enabled high packing densities for both green and sintered body of printed Al (1.55 % and 0.30% void ratios, respectively). The optimized packing then led to the achieved good mechanical strengths (~30 MPa in compressive strength) for Al objects, which were comparable to or even higher than the ones printed by other sophisticated metal 3DPs based on high powered lasers (**Supplementary Fig. 18**). The porosity of the green body, on the other hand, still has a room for enhancements, even though the value (41 %) itself is well within the previous reports (30-60 %) ⁶⁰⁻⁶². Furthermore, the resolution of the 3D printed objects, which is another important aspect along with mechanical strengths, can be enhanced by controlling the evaporation of jetted water. The inevitable liquid-powder imbibition could be reduced by elevating the builder temperature near evaporation point (70 °C) of water, of which the plausible insufficiency in the metal bridging from the fast evaporation was compensated by using an additional step of the humification which completes the remnant chelation reactions between metal particles and citrates (**Supplementary Fig. 19**). The whole printing process based on citric-acid-based BJM3DP can be completed through the thermal treatments of green bodies, where the debinding and sintering result in the merging of metal particles and thus provides an abrupt enhancement in the mechanical strengths. In order to minimize the rapid shrinkage and maintain the structural integrity during the initial debinding process, the optimization of debinding temperature and duration is required to remove an appropriate amount of chelation bridges at mild rate. Since the citric acid thermally decomposes above 175 °C, which is far lower than that of other water-soluble polymeric binders such as PVP, it is necessary to choose lower debinding temperature. A fine adjustments and optimizations are out-of-scope for this work and would require further study, but our choice of conditions (350 °C for 3 hours) did not sacrifice the structural integrity. This work also demonstrated that, the chelation chemistry of citric acid observed in Al could be expanded to various metals such as Cu, Fe and Ti-Al6-4V alloy. Based on these aspects, the proposed facile approach of using environmentally friendly chelators is expected be a cornerstone for promoting the development of the highly approachable, low-cost and safe consumer-level desktop metal 3D printing systems.”

15. Methods: *The methodologies were not described in detail, with incomplete descriptions (for instance heat treatment).*

Reply: Thank you for the comment. We have moved the Methods section into the revised Supplementary Information and revised it to include more detailed descriptions of the experiments.

Materials.

Supplementary Information, Line 32-36: “...and sodium citrate ($\geq 99\%$, Sigma-Aldrich). Al powder was purchased from Henan Yuanyang Powder Technology Co., Ltd (FLPN10 for 10 μm), from Korea Powder Co., (CAS#: 7429-90-5 for 75 μm powders) and from Changsung (≤ 40 mesh, 355 μm). The size distribution was confirmed by using ISO 13320 laser diffraction method (Beckman Coulter LS-13-320), as presented in Supplementary Fig. 16. Then, each chelator powder was uniformly mixed at a concentration of 20% (w/w) with the metal powder and sieved through a mesh.”

BJM3DP process.

Supplementary Information, Line 48-49: “...evaporation rate of water during the 3D printing process. The process parameters are summarized in the Supplementary Table 2.”

Supplementary Information, Line 47: “...fixed at 2000 mm min^{-1} . The temperature of both the builder platform and the inkjet nozzle was maintained at 70 $^{\circ}\text{C}$ to control the evaporation rate of water during the 3D printing process.”

Supplementary Information, Line 51-55: “...under vacuum (0.03 torr). The heat treatment profiles are shown in Supplementary Fig. 20. 3 cycles of the following heat treatment profile are conducted: (Debinding) Heating from RT to 350 $^{\circ}\text{C}$ by 10 $^{\circ}\text{C}/\text{min}$ followed by 350 $^{\circ}\text{C}$ treatment for 3 hours. (Sintering) Heating from 350 $^{\circ}\text{C}$ to 620 $^{\circ}\text{C}$ by 10 $^{\circ}\text{C}/\text{min}$ followed by 620 $^{\circ}\text{C}$ treatment for 15 hours. (Cooling) Cooling from 620 $^{\circ}\text{C}$ to RT by -10 $^{\circ}\text{C}/\text{min}$.”

Characterization.

Supplementary Information, Line 61-64: “...with a compression rate of 8 mm s^{-1} . The bulk density and porosity of printed objects were acquired by using 3D computed tomography (μCT , Nikon XTH 320) with the voxel size of 1 μm at X-ray beam energy of 210 kV. Voxel analysis and 3D visualization were performed by using VGSTUDIO MAX (Volume Graphics Pte. Ltd).”

Supplementary Fig. 20. The heat treatment profile used for the debinding and sintering of Al-based 3D printed objects.

Parameter	Value	Unit
Powder spreading speed	1000	mm/min
Printing speed	2000	mm/min
Ink droplet size	33	pL

Printing resolution	600	DPI
Inkjet nozzle temperature	70	°C
Builder platform temperature	70	°C

Supplementary Table 2. Summary of 3D printing process parameters.

REVIEWERS' COMMENTS

Reviewer #2 (Remarks to the Author):

The responses to the reviewers' comments are satisfactory.

Reviewer #3 (Remarks to the Author):

The authors were highly dedicated and motivated to significantly improve the quality of the manuscript. The authors have diligently worked to address the reviewer concerns, as they were stated. The work is interesting and well-supported with data, but there is still a challenge on my end to justify the presence of the work in this journal, when comparing to the other body of knowledge in the field. The combined manuscript and supplementary data could make for a well-rounded journal article elsewhere. From an editorial perspective, there are only minor recommendations:

1. Introduction: page 2, line 48. "Binder jetting 3D printing (BJM3DP)", do the authors mean to insert "metal" for "M" in the nomenclature?
2. Results: Fig 1a, "metal powder" could be replaced with "metal powder and chelator"
3. Results: page3, the authors should mention Al powder was used before mentioning the hazards of such in page 4, line 106-112. Perhaps adding a statement after page 3, line 92 "Next, two gantry boxes were filled with the mixture of the metal powder and chelator; for the purpose of this work, Al powder was utilized.
4. Discussion: define "void ratio"; to my knowledge, it is not reported as %. The use of the void ratio is also a bit confusing, as it is a more common approach to report porosity (%) instead.
5. It is recommended that the authors follow the same structure as the other papers in Nature Communications in terms of header sections. These do not seem to align (for instance, the Methods header is missing).

Point-by-point responses to the reviewer's comments are listed below:

Reviewer #2 (Remarks to the Author):

The responses to the reviewers' comments are satisfactory.

Reviewer #3 (Remarks to the Author):

The authors were highly dedicated and motivated to significantly improve the quality of the manuscript. The authors have diligently worked to address the reviewer concerns, as they were stated. The work is interesting and well-supported with data, but there is still a challenge on my end to justify the presence of the work in this journal, when comparing to the other body of knowledge in the field. The combined manuscript and supplementary data could make for a well-rounded journal article elsewhere. From an editorial perspective, there are only minor recommendations:

1. Introduction: page 2, line 48. *“Binder jetting 3D printing (BJM3DP)”, do the authors mean to insert “metal” for “M” in the nomenclature?*

Reply: Thank you for the comment. We meant to insert “metal” for “M” in the nomenclature. We have replaced “Binder jetting 3D printing (BJM3DP)” with “Binder jetting metal 3D printing (BJM3DP)”.

Page 2, Line 50: “**Binder jetting metal 3D printing (BJM3DP)** is a promising AM technique that ...”

2. Results: Fig 1a, *“metal powder” could be replaced with “metal powder and chelator”*

Reply: Thank you for the comment. We have replaced the term “metal powder” with “metal powder and chelator” on Figure 1a.

Figure 1a: “**metal powder and chelator**”

3. Results: page3, *the authors should mention Al powder was used before mentioning the hazards of such in page 4, line 106-112. Perhaps adding a statement after page 3, line 92 “Next, two gantry boxes were filled with the mixture of the metal powder and chelator; for the purpose of this work, Al powder was utilized.*

Reply: Thank you for the valuable comment. We believe that expressions were not well formulated and as a result it did mislead the reviewers. Therefore, we added a provided statement in the revised manuscript as follows (highlighted in yellow):

Page 3, line 95: “Next, two gantry boxes were filled with the mixture of the metal powder and chelator; **for the purpose of this work,** Al powder was utilized. Each of these gantry boxes ...”

4. Discussion: define “void ratio”; to my knowledge, it is not reported as %. The use of the void ratio is also a bit confusing, as it is a more common approach to report porosity (%) instead.

Reply: Thank you for the valuable comment. We used the term “void ratio” as the value shown in the CT measurement result was taken as it is. The porosity obtained by mercury intrusion porosimetry has already been mentioned in the manuscript, but if the defect volume ratio (%) obtained from the CT results is expressed in the same porosity, it seems that confusion may arise about the two results, so the term (defect volume) of CT data ratio) was used We believe that expressions were not well formulated and as a result it did mislead the reviewers. Therefore, we corrected the corresponding part in the revised manuscript as follows (highlighted in yellow):

Page 6, Line 204: “A good packing density (defect volume ratio of 4.92 %) of the green body based on coarse Al powder (<355 μm) by ...”

Page 7, Line 212: “...with fine particles (Supplementary Fig. 15 and 17) thus led defect volume ratios to be critically reduced (1.55 % and 0.30 % for green and sintered body, respectively).”

Page 8, Line 268: “...for both green and sintered body of printed Al (1.55 % and 0.30 % defect volume ratios, respectively). The optimized packing then ...”

Figure 3d: “defect volume ratio: 1.55%”

Figure 3d legend: “d Rendered image and defect volume ratio from μCT analyses for green body based on metal particles with ...”

5. It is recommended that the authors follow the same structure as the other papers in Nature Communications in terms of header sections. These do not seem to align (for instance, the Methods header is missing.

Reply: Thank you for the valuable comment. We corrected the corresponding part in the revised manuscript as follows, referring to ‘Nature Communications Guide to Formatting Articles’ :

1. We moved the supplementary methods to the main text under the ‘Methods’ heading. (page 9-10, Line 292-329)